# Relation of Fruits and Vegetables with Major Cardiometabolic Risk Factors, Markers of Oxidation, and Inflammation

**DOI:** 10.3390/nu11102381

**Published:** 2019-10-06

**Authors:** Maria Lapuente, Ramon Estruch, Mana Shahbaz, Rosa Casas

**Affiliations:** 1Department of Internal Medicine, Hospital Clinic, Institut d’Investigació Biomèdica August Pi i Sunyer (IDIBAPS), University of Barcelona, Villarroel, 170, 08036 Barcelona, Spain; calabardina.maria@gmail.com (M.L.); restruch@clinic.cat (R.E.); mana.shahbaz@gmail.com (M.S.); 2CIBER 06/03: Fisiopatología de la Obesidad y la Nutrición, Instituto de Salud Carlos III, 28029 Madrid, Spain

**Keywords:** fruit and vegetables, noncommunicable diseases, chronic diseases, bioactive compounds, immune system, inflammation

## Abstract

Noncommunicable diseases (NCDs) are considered to be the leading cause of death worldwide. Inadequate fruit and vegetable intake have been recognized as a risk factor for almost all NCDs (type 2 diabetes mellitus, cancer, and cardiovascular and neurodegenerative diseases). The main aim of this review is to examine the possible protective effect that fruit and vegetable consumption or their bioactive compounds may have on the development of NCDs such as atherosclerosis. The accumulated evidence on the protective effects of adequate consumption of fruits and vegetables in some cases, or the lack of evidence in others, are summarized in the present review. The main conclusion of this review is that well-designed, large-scale, long-term studies are needed to truly understand the role fruit and vegetable consumption or their bioactive compounds have in atherosclerosis.

## 1. Introduction

The principal cause of death worldwide is cardiovascular disease (CVD), being the leading cause of death in developed countries [1]. The World Health Organization (WHO) estimated 17.7 million deaths were due to CVD in 2015, representing 31% of all causes of death [2]. 

Atherosclerosis is the main cause of CVD. It is a chronic, generalized, and progressive disease, which results from a chronic inflammatory process that affects the arteries of different vascular beds and is characterized by thickening of the intimate layer and a loss of elasticity in half of the cases [3]. Both oxidative stress and systemic inflammation can be modified by a healthy lifestyle, including a healthy and balanced diet, physical activity, moderate alcohol consumption, and stopping tobacco use [4,5,6].

On the other hand, according to the WHO, in 2017, an estimated 3.9 million deaths worldwide were attributable to inadequate fruit and vegetable (F/V) consumption [7]. Moreover, noncommunicable diseases (NCDs) such as type 2 diabetes mellitus (T2DM), cancer, and heart disease are collectively responsible for over 70% of all deaths worldwide, that is, 41 million people [8]. Indeed, all NCDs are known to present low-grade inflammation that characterizes these pathologies with high concentrations of some inflammatory biomarkers.

It is known that F/V are good sources of dietary fiber, vitamins, minerals, and many non-nutrient substances that are beneficial for our health, such as flavonoids, plant sterols, and other antioxidants [7]. Taking into account the nutritional composition of F/V, they should make up a large portion of our diet. In fact, the WHO and other health organizations recommend increased intake of F/V (≥400 g/day) to improve overall health and reduce the risk of NCDs, which include illnesses such as heart disease, cancer, diabetes, and obesity, as well as for the prevention and alleviation of several micronutrient deficiencies [7]. 

Adequate F/V intake as a part of a healthy dietary pattern has been demonstrated to reduce the prevalence of the most frequent NCDs such as T2DM, CVD, obesity or metabolic syndrome (MetS), which are risk factors of CVD [9]. This is due to the potential properties of the phytochemical content of F/V (Figure 1) such as antioxidation, inflammatory biomarkers modulation, anti-platelet and anti-aggregation, as well as improvement of the lipid profile, glucose metabolism, and blood pressure [10,11]. Thus, an adequate intake of F/V as part of the daily diet may reduce the risk of some NCDs including CVDs, certain types of cancer [7], and all-cause mortality [12,13]. Inverse associations have also been observed between the intake of F/V separately and combined and the risk of coronary heart disease (CHD), stroke [14], and CVD [12]. 

According to Scicchitano et al. [15], to date, there is vast scientific evidence about the effectiveness of nutraceuticals against the development of CVD and in reducing the burden of the atherosclerotic process. Nevertheless, how these nutraceuticals exert their positive action on the cardiovascular system is not yet known. It should be highlighted that the positive health effects of these bioactive compounds have mainly been based on observational, in vitro, and in vivo studies [15]. While there is sufficient in vitro evidence supporting the antioxidant, anti-inflammatory, and antidiabetic effects, among others, of allicin, flavonoids, or stilbenes, the in vivo studies available have reported controversial results, indicating that further research are required. To demonstrate and corroborate the results observed in these in vitro and in vivo studies, large controlled clinical trials are needed in humans [15,16,17,18,19]. To know the underlying mechanisms involved in the absorption of different biocompounds according to the health status of the individual, ethnicity, sex, or age, as well as the possible bioavailability, degradation, metabolism, and excretion of this individual should be studied since this might partly explain the differences observed among studies performed. In this context, and according to Castro-Barquero et al. [20], gut microbiota plays a key role in the metabolization of dietary polyphenols. Gut microbiota can modify the absorption, bioavailability, and biological activity of dietary polyphenols, which may explain the contradictory results obtained among the different studies.

In this article, a bibliographic review was carried out using the PubMed, Science Direct, Scopus, and Cochrane Library databases. It was written based on the most relevant articles and studies made in human subjects published no longer than approximately seven years ago and reviewed in the English language literature of humans with no time restriction. The keywords used for this search were fruits and vegetables, atherosclerosis, bioactive compounds, immune system, inflammation, inflammatory markers, oxidative stress, cytokines, phytochemicals, nutraceuticals, etc. The exclusion criteria consisted in a) interventional studies published before 2013; b) articles using juices, smoothies, or similar products; c) articles not containing some of the characteristics mentioned in inclusion criteria; and d) interventions made in animals, ex vivo, or in silico.

The main objective of this review was to perform a bibliographical search to better understand the most recent evidence available correlating F/V consumption with major cardiometabolic risk factors, markers of oxidation, and inflammation. Furthermore, we aimed to investigate and quantify the magnitude of the beneficial effects observed and analyze the potential pathways by which selected food-phytochemicals exert their positive effects.

## 2. Dietary Polyphenol and Carotenoid Intake

One point to consider in this review is dietary polyphenol intake. In Mediterranean countries such as Spain, Greece, Italy, or the South of France, the intake of total polyphenols is around 1011 mg/day in comparison with the United Kingdom (1521 mg/day) and non-Mediterranean countries (1284 mg/day) [20]. According to Zamora-Ros et al. [21], flavonoids (49–62%) represent the highest intake of total polyphenols in Mediterranean countries, followed by phenolic acids (34–44%), lignans (0.3–0.5%), and stilbenes (0.1–0.5%). On the other hand, tomatoes and tomato-based products make up the highest contribution to dietary lycopene intake (15–55%) in European countries [22]. Fruits are the main food source of flavonoids (around 45%), while wine is the main source of stilbenes [21].

On one hand, with regard to the habitual intake of polyphenols, it has been reported that total polyphenol intake is 664 mg/day, of which the weighted dietary mean intake of phenolic acids is 363 mg/day, being 259 mg/day for flavonoids [20]. The dietary intake of anthocyanins has been estimated to be around 11.6 mg/day for individuals aged ≥20 years [23]. The average range of flavan-3-ol intake (a subclass of flavonoid) was also estimated to be 180 to 600 mg per person per day [24], while the mean intake of proanthocyanidins is around 95 mg/day in the American population [25], and stilbenes intake, as a sum of resveratrol and piceid, would be between 0.933 to 4000 µg per person and day [26].With respect to hydroxytyrosol, its estimated consumption per day in Spain has been estimated at 5.6 mg (0.3 mg from extra virgin olive oil (EVOO) and 5.3 mg from olives) considering an average consumption of EVOO and olives of 15 g/day and 7 g/day, respectively [27]. Hydroxytyrosol has a bioavailability of 99%, making it easily is integrated into our organism. Therefore, it has been reported that a daily intake of 5.6 mg of hydroxytyrosol could exert great benefits for the human body [27].

On the other hand, Burrows et al. [28] reported that the mean intake of the most common dietary carotenoid, lycopene, is of 4555.4 µg/day. Finally, there is no standard available for the intake of garlic. On one hand, it has been suggested that health benefits can be observed after a daily intake of 1–2 cloves garlic or around 4 g of whole garlic, but there is no scientific reference to support this [29]. Other authors have reported that an average daily dose of dehydrated garlic powder of 900 mg/day has health benefits, although according to several studies the effectiveness of aged garlic extract (AGE) on immune enhancement in humans varies from 1.8 to 10 g per day [29].

Finally, Sones et al. [30] also estimated the average daily intake of total and individual glucosinolate content in both the fresh and cooked cruciferous vegetables using data from the 1980 National Food Survey in the UK. Thus, they reported that the mean daily intake of glucosinolates was 6.7 mg/day of oxazolidine-2-thiones and 17.7 mg/day of thiocyanate ion. Although the estimated daily intake of total glucosinolates was 75 mg/day, the authors also reported that certain individuals could have consumption higher than 300 mg total glucosinolates per day.

## 3. Health Potential of Selected Fruits and Vegetables

While multiple factors can contribute to the incidence and prevalence of atherosclerosis, the prevention of this disease can be reduced by the high consumption of F/V. Of all the possible foods and phytochemicals involved, we have selected those showing the greatest evidence of having a role in the prevention of atherosclerosis in the last years (Figure 1).

### 3.1. Tomatoes

The tomato fruit belongs to the Solanaceae family and is one of the most economically important and most frequently produced plants in terms of food production [31]. Tomatoes contain a great number of phytochemicals, which include a wide range of vitamins, minerals, and antioxidants. There is a large amount of scientific evidence on the beneficial effects of tomato fruit on chronic NCDs, including CVD, hypertension, T2DM, obesity, and cancer [32,33]. 

Tomatoes present a high content of water (93–95%) and are made up of 3% of carbohydrates, 1.2% of proteins, and 1% of total lipids. Their vitamin content is important, especially vitamins A, E (mainly α-tocopherol), C, and folate, as well as minerals such as potassium and iron, among others [32,33]. Absorption of iron is favored by the presence of ascorbic acid (vitamin C) which, when present in sufficient quantity, improves the bioavailable of iron [33,34]. Thus, tomato juice can provide around 10–20% of the recommended daily allowance of iron [34]. Other acids also present in tomatoes include succinic, oxalic, and tartaric acid, but these only constitute a small fraction compared to citric acid [34]. In addition, they are an excellent source of bioactive compounds such as phenolic compounds (phenolic acids and flavonoids) and carotenoids (lycopene, β-carotene, lutein, and zeaxanthin) [32,33,34,35]. It is of note, however, that the variety of tomato cultivated, the grade of ripening at harvest, irrigation system, light, temperature, air composition, etc. can affect the chemical composition of tomatoes [32].

Phenolic compounds plus vitamin C, the hydrophilic fraction, confer the highest total antioxidant activity of ripe tomato (83%), while the lipophilic fraction (carotenoids, vitamin E, and lipophilic phenols) represents only 17% of the total [36]. The red color of tomatoes is due to their carotenoid compounds. Lycopene is the main carotenoid found in significant amounts in tomatoes, followed by β-carotene [33,36].

The most prominent beneficial feature of tomatoes is their antioxidant effects because of their content of carotenoids (mainly, lycopene) and antioxidant vitamins such as vitamin A, C, or E [33,37,38]. Indeed, lycopene seems to play an important role in the prevention of different NCDs (CVD and several types of cancer) thanks to the high number of double bonds present in its structure, which increase its capacity to suppress reactive oxygen species (ROS) and hypochlorous acids [33,35,39]. It is estimated that the effectiveness and antioxidant activity of lycopene is 10 times higher than that of vitamin E [40]. 

Several epidemiological studies and meta-analyses have reported a dose-response relationship between serum lycopene levels and the risk of CVD and major risk factors of CVD (atherosclerosis, T2DM, hypertension, inflammation, and oxidative stress) [41,42,43,44,45,46].

The protective cardiovascular effects of lycopene can be explained by several mechanisms that include improvement of the metabolic profile (lipid and glucose metabolism), intercellular communication and arterial stiffness, significant reductions of levels of inflammatory response, oxidative stress markers and lipid peroxidation, as well as, antiatherosclerostic, antiplatelet, anti-apoptotic response, antioxidant, and antihypertensive effects and protective endothelial effects (nitric oxide bioavailability and blood flow) [46,47,48].

In a recent systematic review and meta-analysis including 22 publications based on interventional studies in humans, Cheng et al. [49] reported that consumption of tomato-based products or lycopene supplements led to a significant reduction of the serum concentrations of low-density lipoprotein cholesterol (LDL-c) (−0.22 mmol/L; *p* = 0.006), systolic blood pressure (SBP) (−5.66 mmHg; *p* = 0.002), and improved the endothelial function measured as flow-mediated dilation (FMD, 2.53%; *p* = 0.01) and inflammatory markers such as interleukin (IL)-6 (standardized mean differences (SMDs): −0.25; *p* = 0.03). Nevertheless, it should be noted that lycopene supplements were not able to reduce LDL concentrations. Cheng et al. [49] also reported that diastolic blood pressure (DBP), high-density lipoprotein (HDL)-cholesterol and triglyceride (TG) concentrations, as well as inflammatory (C-reactive protein (CRP)) and adhesion markers such as intercellular adhesion molecule-1 (ICAM-1) improved, albeit not significantly, with a high intake of lycopene-containing food or supplements. 

Several epidemiological studies (Table 1) have reported significant reductions of inflammatory biomarkers such as monocyte chemoattractant protein-1 (MCP-1) [50], vascular adhesion molecule 1 (VCAM-1) [51], IL-6, IL-8 and tumour necrosis factor (TNF)-α, after the intake of tomato juice [52]. Valderas-Martinez et al. [53] also reported significant reductions of MCP-1 and a significant increase in IL-10 concentrations after the consumption of raw tomato or tomato sauce with refined olive oil (TSOO). These authors reported that TSOO intake led to significant reductions in IL-6, IL-18, and VCAM-1 concentrations, as well as significant reductions in the leukocyte expression of lymphocyte function–associated antigen (LFA) in T-lymphocytes and CD36 in monocytes. On the other hand, Thies et al. [54] failed to find a significant improvement in the biomarkers analysed (oxidized LDL (oxLDL), ICAM-1, and IL-6) after a high daily consumption of tomato-based products (equivalent to 32–50 mg lycopene) or lycopene supplements (10 mg).

Other interventional studies have described lycopene anti-oxidative effects. Burton-Freeman et al. [55] reported that tomato products are able to significantly reduce oxLDL (*p* < 0.05) after consuming high-fat meals during two weeks. Nevertheless, these authors reported a significant rise in IL-6 concentrations (*p* < 0.0001). Deplanque et al. [56] also showed significant reductions in oxLDL concentrations (*p* < 0.0001) after the intake of a carotenoid-rich tomato extract during two weeks. On the other hand, García-Alonso et al. [51] reported that the intake of tomatoes or tomato products increased the resistance of LDL to oxidation. Ghavipour et al. [57] also found that tomato juice intake could reduce oxidative stress, increasing erythrocyte antioxidant enzymes and plasma total antioxidant capacity in overweight females after daily consumption of tomato juice. Nonetheless, no significant changes were observed in antioxidant biomarkers (superoxide dismutase, glutathione peroxidase, and catalase) after daily consumption of tomato juice (60 mg lycopene) for 20 days [58].

In relation to the anti-platelet effects of tomatoes, a recent double-blinded randomized controlled trial (RCT) showed that consumption of a dietary anti-platelet (water-soluble tomato extract) suppressed platelet function by around one-third of 75 mg of aspirin daily [59] after analyzing 47 healthy volunteers over one week. Similar results were reported by Krasinska et al. [60], who observed significant reductions of 8.6% (*p* < 0.05) of aspirin reaction units after four weeks of treatment with a standard tomato extract. Lazarus et al. [61] also reported that platelet aggregation decreased (area under the curve (AUC), −4963 (3262) vs. −370 (2274); *p* = 0.002) after daily consumption of 250 mL of tomato juice compared with a control group after three weeks. 

The potential beneficial effects of tomato on endothelial function have also been studied. Xaplanteris et al. [62] reported that endothelial function, measured as FMD, was improved by 3.3% and total oxidative status decreased (*p* = 0.038) after daily consumption of 70 g of tomato paste (33.3 mg of lycopene) at midterm (15 days). In contrast, no significant effect was observed on endothelial function after the consumption of 70 g of a buttered roll with and without tomato purée in a cross-over design in healthy non-smoking postmenopausal women [63]. In a similar study with 80 g of tomato paste purèe, the authors concluded that tomato paste had no effect on FMD [64]. 

Despite a large amount of evidence regarding the positive benefits of lycopene on CVD prevention, there are also several studies in humans that did not find any relationship between lycopene consumption and the risk factors of CVD [44,65]. These negative associations might be explained by the bioavailability and metabolism of lycopene, the source of lycopene used (fresh tomato, tomato juice, lycopene supplements, processed tomatoes, etc.), the cardiovascular markers measured (blood pressure, lipid profile, inflammatory markers, final outcomes such as myocardial infarction, stroke or death, etc.), the lycopene doses used (lower to habitual daily intake, 4555.4 µg/day), and the possible interactions with other compounds contained in other foods that can potentiate or inhibit positive lycopene effects, as well as the number of subjects studied that might decrease the statistical power, the length of the intervention, and the type of study performed (prospective, crossover, observational, etc.).

### 3.2. Garlic

Garlic, onions, shallots, leek, and chive belong to the Liliaceae family [66]. Garlic (*Allium sativum*) has been widely used in many cultures for thousands of years for both culinary and medicinal purposes [67]. At present, garlic is one of herbal medicines most frequently investigated and used as complementary therapy because of its potential beneficial effects on health [68].

Garlic is characterized by its high content of sulphur compounds such as diallyl sulphate, alliin or S-allyl-cysteine sulfoxide, ajoene, and allicin compared with other Allium species. The main bioactive compound present in garlic is allicin (up to 1.8% in a garlic bulb) [68,69]. The main compounds responsible for the flavor and odor of garlic are mostly sulphur-containing, non-volatile amino acids (thiosulfinates), such as S-allyl-cysteine sulfoxide [68,69,70]. Moreover, garlic contains several enzymes (e.g. alliinase enzyme) that are sensitive to high temperatures (≥60 °C) and minerals (e.g. selenium). Garlic intake has many beneficial properties that have demonstrated therapeutic and preventive roles in human health. As mentioned above, sulphur compounds seem to be responsible for most of these beneficial effects. On one hand, several in vitro studies have demonstrated its anti-bacterial, anti-viral, and anti-fungal effects [70,71,72]. Several clinical trials have also reported significant actions of fresh garlic on cardiovascular risk factors [66,69,70,73,74], as well as anti-inflammatory [67,73,74,75] and anti-microbial effects [71,72,73] and anti-cancer properties [70,71,72,76]. In fact, a large amount of scientific evidence has demonstrated the efficacy of garlic in the treatment and prevention of atherosclerosis due to its anti-atherogenic properties [75].

Several meta-analyses have reported that AGE induces anti-hyperlipidemia by reducing the level of total cholesterol and LDL-c [77,78,79]. AGE contains potent unique compounds such as S-allyl-cysteine sulfoxide [71]. Although a recent systematic review and meta-analysis performed by Shabani et al. [79] found a significant improvement in the levels of HDL-c (9.65mg/dL, *p* = 0.001) and TG (12.44 mg/dL, *p* = 0.001) concentrations, these effects were not observed in other studies [72,73]. In relation to T2DM, AGE supplementation also showed an anti-diabetic effect by significantly reducing glucose parameters such as fasting blood sugar and glycosylated hemoglobin (HbA1C) [79,80].

Garlic supplementation has also shown to have beneficial effects on hypertension. Several meta-analyses have reported significant reductions of SBP and DBP in hypertensive individuals [81,82,83].

In relation to anti-atherogenic effects, garlic has shown to have anti-inflammatory and antioxidant effects. A recent meta-analysis including nine RCTs (Allium supplement: ≥1200 mg/day or <1200 mg/day) reported significant reductions in the CRP levels (weighted mean difference (WMD): −0.8 mg/L, 95% confidence interval (CI): −1.5, −0.1, *p* = 0.02) [84]. Similar results (12 to 3600 mg/day of AGE, 16 RCTs) were described by Darooghegi et al. [85], who reported significant changes in CRP concentrations (WMD: −0.61 mg/L, 95% CI: −1.12, −0.11, *p* = 0.018). The same meta-analysis also described significant reductions of IL-6 (WMD: −0.73 ng/L, 95% CI: −1.06, −0.40, *p* < 0.001), and TNF-α (WMD: −0.26 ng/L, 95% CI: −0.41, −0.12, *p* < 0.001) compared to the control group.

An RCT including 42 volunteers undergoing peritoneal dialysis (Table 2) performed by Zare et al. [86] found significant reductions of CRP concentrations (−71.4%) and IL-6 (−51.7%) after the consumption of 400 mg of standardized garlic Allium (twice a day) for two months. Kumar et al. [87] also found significant reductions in CRP levels after the daily administration of 500 mg of garlic (Allium sativum) capsules. In another RCT, in which volunteers were supplemented with 1200 mg of AGE plus 120 mg of CoQ10 daily, significant reductions in CRP levels were observed compared to the control group (−0.12 ± 0.24 vs. 0.91 ± 0.56 mg/L, *p* < 0.05) [88]. Nonetheless, the RCTs performed by Atkin et al. [89] and Williams et al. [90] showed no significant changes in the CRP concentrations or IL-6 after administering 1200 mg/day for four weeks or 2400 mg/day for 6 weeks an AGE to diabetic volunteers, respectively. Neither did Ried et al. [70] find significant changes in the TNF-α and IL-6, concentrations although it was effective in lowering BP and improved arterial stiffness in patients with uncontrolled hypertension (*p* < 0.05; both). Furthermore, AGE improved gut microbiota, increasing the levels of Lactobacillus and Clostridia species after three months of supplementation with 1.2 mg of S-allylcysteine. On the other hand, more recently, after six months of black garlic intake, a significant reduction of brain natriuretic peptide (BNP), a marker of heart failure (HF), was also reported, further increasing circulating antioxidant levels [91].

Very few clinical trials have studied the possible association between garlic consumption and oxidative stress. One of these studies was carried out by Wang et al. [92], who reported significant reductions of markers of oxidative stress, all related to atherosclerosis, including reactive oxygen metabolites (dROMs), lipid peroxide, and 8-iso prostaglandin F2α concentrations after the consumption of black garlic for 14 days. Black garlic has a high concentration of polyphenols and 10-fold stronger antioxidant properties, tetrahydro-β-carboline derivatives, organosulfur compounds such as S-allyl-cysteine and S-allyl-mercaptocysteine, and flavonoids compared with fresh garlic [92]. 

Based on current research, garlic has shown favorable effects against atherosclerosis, suggesting that it might be useful as an adjuvant therapy for vascular diseases. Nevertheless, the lack of evidence and the confounding results obtained in some clinical trials make it necessary to perform more large, long-term clinical trials, including the exact dose of bioactive compounds, the type of garlic used, a sufficient sample size, the measurement of similar cardiovascular markers, and the inclusion of healthy or non-healthy individuals, etc. Indeed, up to now, there is no consensus regarding what dose should be used to achieve immune enhancement in human because the dose of AGE used varies widely (1.8–10 g/day).

### 3.3. Berries

Berries are the term used in this chapter to include fruits such as strawberries, raspberries, blueberries, black currants, and blackberries, among others [93,94]. Berries are characterized as being low in calories and having a high content of vitamins such as vitamins A, B complex, C, E, essential minerals such as calcium, magnesium, selenium, etc., as well as fat (especially unsaturated) and dietary fiber such as pectin (soluble fiber) [93,94]. Furthermore, compared with other foods eaten, berries, have the highest antioxidant capacity because their high content of antioxidant compounds such as polyphenols (procyanidins, quercetin, phenolic acids, and anthocyanins mainly), carotenoids, and vitamin C [93,94,95]. It is of note that the amount of antioxidant components varies between berry species, as well as the climatology, the grade of ripening, post-harvest treatments and processing, etc., with each berry having a very different nutritional or phytochemical profile with respect to others [93]. It has been estimated that the average polyphenol content of black currants, raspberries, and strawberries is approximately 300–100 mg/100g [93]. 

There is increasing scientific evidence shows that berry intake is closely associated with the prevention of CVD through several mechanisms such as improving the lipid profile, inflammation, T2DM, hypertension, and coagulation [93,94,95,96,97,98]. For example, the consumption of 1–2 portions of strawberries, raspberries, and blueberries daily is associated with a lower risk of CVD [96]. This protective effect seems to be attributed to polyphenols, which are considered to be bioactive compounds found in blueberries [99].

To date, there is still scarce evidence (clinical trials) on the molecular mechanisms by which the bioactive compounds exert their health benefits [94]. Pomegranates, blueberries, strawberries and acai are the berries most frequently studied, due to their preventive effects on atherosclerosis [87,88]. All of these berries seem to have similar pathways of action mainly based on their anti-inflammatory and antioxidant activity [99,100,101]. Several epidemiological, clinical, and animal studies have indicated diverse health benefits of anthocyanins, a type of polyphenol (flavonoids), responsible for the color of berries (from red to purple to blue) [93,94]. These benefits in the prevention of atherosclerosis or other cardiovascular complications include improvement of endothelial function due to the activation of endothelial nitric oxide synthase (eNOS) signaling, reducing oxidative stress and modulating inflammation response, as well as improving the lipid profile [98].

A recent meta-analysis [95] including 22 RCTs and 1251 subjects focused on the effect berry intake has on CV risk factors and reported improvements in the levels of LDL-c (WMD: −0.21 mmol/L, 95% CI: −0.34, −0.07, *p* = 0.003), SBP (WMD: −2.72 mmHg, 95% CI: −5.32, −0.12, *p* = 0.04), fasting glucose (WMD: −0.10 mmol/L, 95% CI: −0.17, −0.03, *p* = 0.004), body mass index (BMI) (WMD: −0.36 kg/m2, 95% CI: −0.54, −0.18, *p* < 0.00001), glycated hemoglobin (HbA1c) (WMD: −0.20%, 95% CI: −0.39, −0.01, *p* = 0.04), and TNF-α (WMD: −0.99 pg/mL, 95% CI: −1.96, −0.02, *p* = 0.04). Similar results were reported in another systematic review and meta-analysis performed by Yang et al. [102] that included 32 RCTs and 1491 participants. They observed significant reductions in fasting glucose (SMD: −0.31; 95% CI: −0.59, −0.04; I^2^ = 80.7%), 2-h postprandial glucose (SMD: −0.82; 95% CI: −1.49, −0.15; I^2^ = 77.7), HbA1c (SMD: −0.65; 95% CI: −1.00, −0.29; I^2^ = 72.7%), total cholesterol (SMD: −0.33; 95% CI: −0.62, −0.03; I^2^ = 86.9%), and LDL (SMD: −0.35; 95% CI: −0.66, −0.05; I^2^ = 85.2%). The benefits observed in cardiometabolic disease where attributed to the action of anthocyanins.

Furthermore, anthocyanin supplementation has been associated with an improvement of inflammatory status. A systematic review and meta-analysis of 17 RCTs [103] reported significant reductions in the levels of TGs, LDL-c and apolipoprotein B, as well as a significant increase in the HDL-c and apolipoprotein A levels after anthocyanin supplementation (80–500 mg/day for 4h–45 days). The authors also described significant reductions in the levels of inflammatory markers such as TNF-α (mean difference (MD): −1.98, 95% CI: −2.40 to −1.55 pg/mL, I^2^ = 0%, *p* = 0.975). However, the effect of anthocyanin on IL-6 and high sensitivity (hs)-CRP levels was not statistically significant. In another systematic review and meta-analysis of seven RCTs (3–12 weeks, and dose of anthocyanins from 45.1 to 640 mg/day) [104] did not find any significant relationship between anthocyanin dose and CRP concentrations (WMD = 0.018, 95% CI −0.44, 0.47, *p* = 0.94). Neither did the authors find any association between the duration of the trial (slope: 0.01, 95% CI −0.002, 0.03, *p* = 0.08) and anthocyanin dosage, although they did find a significant association between the dose of anthocyanins and CRP levels (slope: 0.001, 95% CI 0.0007, 0.002, *p* < 0.001). Finally, Bloedon et al. [105] studied the impact of anthocyanin-rich fruit, including the type and amount of fruit, as well as the processing methods, on inflammatory and oxidative stress markers. They described significant changes in IL-6, TNF-α, hs-CRP, IL-1 receptor antagonist (IL-1ra) and IL-10 levels as markers of inflammation, and changes in malondialdehyde (MDA) and protein carbonyls levels as markers of oxidative stress. Black currants, tart cherries, and blueberries showed the highest anti-inflammatory effect, while acai and tart cherries showed the highest antioxidant effect.

A randomized, double-blinded, clinical study performed by Lee et al. [106] found significant reductions of MCP-1 and TNF-α concentrations after daily administration of 2.5 g of anthocyanin-rich black soybean testa extracts in 32 overweight or obese participants for eight weeks (Table 3). Zhang et al. [107] also found significant reductions in the levels of several chemokines such as CXCL7, CXCL5, CXCL8, CXCL12, and CCL2 compared with placebo after the administration of 320 mg of anthocyanin (blackcurrant) daily for 6-months to 73 hypercholesterolemic participants. Moreover, significant reductions were also observed for hs-CRP, IL-1β and sP-selectin. Soltani et al. [108] reported significant improvements in the lipid profile and MDA levels compared to the placebo group; however, no significant changes were observed for HDL-c or hs-CRP concentrations. On the other hand, significant improvements in the lipid profile and significant reductions of MDA (−31.40%) and urinary T2DM (8-OHdG, −29.67%) and isoprostane (−27.90%) levels were observed after 23 healthy individuals consumed 500 g of strawberries daily for 30 days [109]. In a study including 42 overweight smoker subjects, Davinelli et al. [110] showed significant reductions in oxLDL and 8-iso-prostaglandin F2α after daily intake of an extract of maqui berry (162 mg anthocyanins) after 40 days of intervention. Furthermore, the plasma total antioxidant capacity was also significantly increased (+24.97%), and the number of activated platelets and spontaneous and oxidative hemolysis were significantly reduced. Nonetheless, no changes in CRP levels were observed after administering 320 mg of anthocyanin (blackcurrant) daily to 80 participants with prediabetes or early untreated diabetes for three months [111].

The controversial results obtained among the different studies presented could be due to an insufficient daily dose of anthocyanin consumed (less than 600 mg/day). In addition, the study duration or the healthy status of the subjects, as well as the sample size used or the source of anthocyanin consumed (whole-food, an extract, etc.) could explain the differences found. Future lines of investigation should be aimed at obtaining more evidence about which mechanisms, anthocyanins or other phytochemicals present in berries exert their positive benefits against vascular disease. It is necessary to identify the bioactive compounds of berries and know the possible the synergies between them. In addition, it should not be forgotten that anthocyanins are metabolized in humans, suggesting that their vascular health effects might be mediated by their metabolites. The metabolites of anthocyanins remain unknown. Therefore, it is necessary to study the molecular mechanisms involved and evaluate the activity of anthocyanin metabolites in vascular health in depth with well-designed, long-term, controlled trials including well established doses of anthocyanins.

### 3.4. Apples

The fruit most widely consumed in several countries are apples, which are consumed both as fresh fruit, juice, or dried [112]. Apples contain several macronutrients and non-nutrient compounds such as dietary fiber, minerals, and vitamins (vitamins C and E, some pro-vitamin A carotenes, lutein, folic acid, potassium, and magnesium) and are also an important source of phytochemicals [113]. Among other phytochemicals, apples contain important antioxidants such as phloridzin, phlorizin, quercetin, catechins, procyanidins, epicatechin, rutin, and chlorogenic acid [114]. Some factors that can affect the content, bioavailability, and quality of the phytochemical compounds include the variety of apple, ripening, storage, and processing [114].

There is growing scientific evidence (clinical, in vitro, and in vivo data) that indicate that the intake of apples and apple products (extracts and juices) may reduce the risk of NCDs (cancer, CVD, asthma, Alzheimer’s disease, T2DM, weight management, bone health, pulmonary function, and gastrointestinal protection) by several mechanisms, due to their antioxidant, anti-inflammatory, antiproliferative, and cell signaling properties [113,115]. In addition, apples and apple products decrease lipid oxidation and improve cholesterol levels. 

Boyer et al. [113] reported that apple products and apple-flavonoid intake, both catechin and epicatechin, were inversely associated with coronary mortality. Finnish women with high daily apple consumption (>71 g) showed significant reductions (43%) in coronary mortality compared to those who did not eat apples, while the reduction in men was 19% (>54 g). 

On one hand, a recent meta-analysis and systemic review [116] including 120 RCTs on the effects of flavanol-contained in different type of foods (tea, cocoa, and apple products) found significant reductions in BMI (standardized difference in means (SDM): −0.15, *p* < 0.001), waist circumference (SDM: −0.29, *p* < 0.001), total-cholesterol (SDM: −0.21, *p* < 0.001), LDL-c (SDM: −0.23, *p* < 0.001), and triacylglycerides (SDM: −0.11, *p* = 0.027), and with an increase of HDL-c (SDM:0.15, *p* = 0.005). Pure apple polyphenols have also been associated with a dose-dependent improvement in total and LDL-c in subjects with mild hypercholesterolemia [117]. Ravn-Haren et al. [118] reported significant reductions of LDL-c after 23 healthy subjects consumed 550 g of whole apple, 22 g of apple pomace or 500 mL of clear and cloudy apple juices, respectively, for one month. Thereby, both polyphenols as the soluble fibre (pectin) could be explained the cholesterol-lowering effect of consumption apples in healthy humans.

Apple intake has also shown to have significant effects on blood pressure and glycemic control [119,120,121]. On the other hand, in seven studies and 587 patients, Serban et al. [119] reported significant reductions both in SBP and DBP after supplementation with quercetin. Analyses by subgroups of studies involving quercetin doses ≥500 mg/day showed significant changes in both SBP (−4.45 mm Hg) and DBP (−2.98 mm Hg), while no changes were observed for quercetin doses less than 500 mg/day. In addition, Borgi et al. [120] studied the incidence of hypertension in three prospective cohort studies (Nurses’ Health Study (*n* = 62,175), Nurses’ Health Study II (*n* = 88,475), and Health Professionals Follow-up Study (*n* = 36,803) and found that the consumption ≥4 servings per week of quercetin was inversely associated with the risk of developing hypertension. Likewise, Ostadmohammad et al. [121] did not observe significant changes in fasting plasma glucose after analysing a pool of ten studies (administering quercetin at a dose of 100–1000 mg/day for ≥4 weeks). Nevertheless, significant changes were reported with quercetin at doses of ≥500 mg/day for ≥8 weeks.

On the other hand (Table 4), in a randomized cross-over study in 20 healthy subjects performed by Soriano-Maldonado et al. [122], the consumption of two glasses of 250 mL/day of apple juice for one month showed significant reductions of inflammatory cytokines such as IL-8, IL-6, IL-10, MCP-1, plasminogen activator inhibitor-1 (PAI-1), E-selectin, VCAM-1, and ICAM-1. CRP levels were also diminished by 22 and 32% after supplementation with dried apple to 160 postmenopausal women for six and 12 months [123]. Furthermore, dried apple significantly decreased plasma lipid peroxide levels by 33% after one year of intervention. Bondonno et al. [124] also reported a higher FMD, lower pulse pressure, and lower SBP (*p* < 0.05; all) after an intervention with dried apple. S-nitrosothiols+othernitrosylated species (RXNO) and nitrite were significantly increased (*p* < 0.05; both) when measured to assess plasma nitric oxide (NO) status. Zhao et al. [125] showed that daily consumption of an apple or supplementation of an apple-polyphenol extract for one month was associated with a drastic reduction in plasma concentrations of oxLDL/beta2-glycoprotein I complex (β2GPI), which has been proposed to lead to the development of atherosclerosis, although the reduction was higher in the case of whole apple intake. None of these apple presentations (consumption of an apple or supplementation of apple-polyphenol extract) were associated with a significant increase of superoxide dismutase (SOD) activity. In contrast, Auclair et al. [126] were unable to demonstrate that consumption of a polyphenol-rich apple improves vascular function FMD.A cross-over study by Saarenhovi et al. [127] did not find significant improvement in FMD and endothelium-independent nitrate-mediated vasodilatation (NMD), markers of inflammation (CRP), adhesion molecules (E-selectin, VCAM-1, and ICAM-1), or coagulation markers (von Willebrand factor (vWF), PAI-1, and asymmetric dimethylarginine (ADMA)) measured after one month of taking an apple polyphenol extract rich in epicatechin and flavan-3-ol oligomers.

Recent evidence-based studies have shown that apple-derived phytochemical compounds seem to play a beneficial role in the prevention of atherosclerosis and vascular damage. However, most of evidence reported is based on human clinical studies in small populations and over a short period of time. Moreover, studies show differences in the plasma markers assessed (lipid profile, metabolism glucose, inflammatory or stress oxidative markers) as well as the polyphenol administered (epicatechin, flavan-3-ol, flavonoids, etc.). It is therefore imperative to establish larger and more extensive clinical studies, with well-defined doses of apple-polyphenol extract, to provide more in-depth information about the possible mechanisms of action of these phytonutrients.

### 3.5. Broccoli

Broccoli as well as brussels sprouts, mustard, cauliflower, radish, cabbage, cress, and kale belong to the Brassicaceae and Cruciferae families [128]. Broccoli contains provitamin A (β-carotene), vitamin C, and vitamin E (tocopherol) and is a good source of phytochemicals due to the high content of carotenoids (lutein, zeaxanthin, β-carotene), phenolic compounds (mainly flavonoids) sulphur glycosides, and minerals (calcium, magnesium, phosphorus, selenium, potassium, and sodium) [129,130,131]. Several studies have reported the potential anticarcinogenic and antioxidant properties of broccoli-rich diets. In fact, higher consumption of glucosinolates (GSL), tocopherols, and carotenoids are associated with a lower risk of cancer, CVD, and poor eye health, respectively [129,132].

GSL are hydrolysed into isothiocyanates and other products such as aliphatic glucoraphanin (GRA) and the indolylic glucosinolates, glucobrassicin, and neoglucobrassicin [129]. Later, glucoraphanin can be hydrolysed by myrosinase, an endogenous plant enzyme or by human intestinal microflora into sulforaphane (SFN) [128,129]. Several animal studies have reported that GRA and especially SFN can act as therapeutic agents in certain forms of cancer (e.g. liver, lung, prostate) because of their capacity to delay or even reverse the development of preneoplastic lesions [128,129].

Carotenoids (lutein, zeaxanthin, β-carotene) and vitamin E (tocopherol) are the lipophilic fraction found in broccoli, and all are characterized by their antioxidant effects [133]. While a high intake of vegetables rich in tocopherols and carotenoids have been associated with a lower incidence of some types of cancer, lutein and zeaxanthin have been associated with a lower incidence of cataracts and macular degeneration [129]. Furthermore, tocopherol has shown to reduce risk of CVD, inhibiting the oxidation of LDL-c [129]. 

It is important to note that the antioxidant capacity of the Brassica and Cruciferae families is controversial, since this capacity can be altered by the cooking method used [134]. For instance, the flavonoid content of broccoli is reduced by 97% by microwave cooking [135]. In contrast, other precooking or cooking methods were not found to alter its antioxidant capacity [136].

Several epidemiological studies have indicated an inverse association and dose–response pattern between broccoli intake and the risk of all-cause and cause-specific mortality CV [137]. In a prospective cohort study including 1226 Australian women aged ≥70 years without clinical atherosclerotic vascular disease (ASVD) followed for 15 years, an inverse correlation was found between cruciferous vegetable intake and ASVD mortality [138]. Indeed, the risk was reduced by 13% (hazard ratio (HR): 0.87; 95% CI, 0.81–0.94, *p* < 0.001)) per a 10 g/day increase.

Several epidemiological studies have studied changes in the lipid profile due to cruciferous vegetable intake. Armah et al. [139] observed an inverse correlation between plasma LDL-c concentrations and a diet rich in high glucoraphanin (HG) broccoli (400 g/week, 21.6 μmol/g GRA). Furthermore, a decrease of plasma LDL levels was higher after consumption of HG broccoli compared to standard broccoli. The authors indicated that the possible mechanism by which glucoraphanin reduces LDL-c is through the induction of nuclear factor erythroid 2-related factor 2 (nrf2)-antioxidant response mediated by SFN derived from glucoraphanin. Bahadoran et al. [140] also reported significant improvements in the lipid profile (TG and total, HDL and LDL-c) and glucose metabolism (fasting glucose) after 72 subjects with T2DM consumed 10 g of broccoli (22.5 μmol/g SFN) daily for four weeks. However, no changes in the lipid profile were described in several RCTs [141,142].

Several clinical studies (Table 5) have reported improvements in inflammatory markers such as CRP and TNF-α after the intake of 10 g/day of broccoli sprout (22.5μmol/g SFN) for four weeks [143]. Likewise, Bahadoran et al. [140,144] reported improvements in the oxLDL/LDL ratio and atherogenic index of plasma, as well as MDA and oxLDL concentrations and the oxidative stress index after daily intake of 10 g of broccoli sprout (22.5 μmol/g SFN) for four weeks. López-Chillón et al. [145] also reported significant reductions of plasma IL-6 and CRP concentrations after consumption of 30 g/day of broccoli sprouts for 10 weeks. A cross-sectional analysis of 1005 middle-aged Chinese women [146] showed that women with a higher consumption of cruciferous vegetables had greater reductions of TNF-α (−12.66%, *p*-trend = 0.01), IL-1β (−18.18%, *p*-trend = 0.02), and IL-6 (−24.68%, *p* trend = 0.02). In contrast no changes or association was found between the consumption of cruciferous vegetables and urinary oxidative stress marker levels (F2-isoprostanes and 2,3-dinor-5,6-dihydro-15-F2t-IsoP, their main metabolite). Finally, Blekkenhorst et al. [147] reported that, for each increase of 10 g of cruciferous vegetable intake, the mean common carotid artery intima-media thickness (CCA-IMT) was reduced in 0.006 mm and in 0.007 mm the maximum CCA-IMT.

Broccoli intake has shown to have potential benefits against mainly CVD and cancer. These benefits seem to be associated with a high GRA and SFN broccoli content. These bioactive compounds might be responsible for decreasing the levels of inflammatory biomarkers related to atherosclerosis progression, as well as markers of oxidative stress. Their consumption is also associated with improvements of glucose metabolism (fasting glucose) and the lipid profile (reducing LDL-c levels). Nonetheless, this evidence is still scarce, needing more well-designed and controlled, long-term interventional studies, which are less heterogeneous and have a large sample size. Most RCTs are short-term with a reduced sample size. It is also important to analyze the dose of these bioactive compounds (the average daily intake of thiocyanate is around 14.7 mg/day) to achieve the maximum benefit and study the possible mechanisms involved in order to obtain more consistent evidence and conclusions.

### 3.6. Cocoa

Cocoa and its main derived products, such as chocolate, have been extensively used for years in many cultures [148]. It is obtained from raw cocoa beans from Theobroma cacao fruits of the cocoa tree [149]. In order to obtain cocoa, several processes are first applied to the raw seeds, which are: fermentation, drying, roasting, winnowing, alkalization, and conching [149].

Cocoa, one of the richest sources of polyphenol and its derived products, contains an important number of phytochemicals, the most important being methylxanthines and flavan-3-ols (including proanthocyanidins) [149]. Moreover, the most characteristic type of methylxanthines found in the Theobroma cacao tree is theobromine (3,7-dimethylxanthine) [149].

Although chocolate has become a food associated with its hedonic properties over the last years, a large amount of scientific evidence suggests that cocoa and dark chocolate can also contribute to beneficial effects on the prevention of some NCDs [150], especially CVD [151], T2DM [152], and hypertension, among others [148,149]. In particular, many clinical and epidemiological studies seem to indicate that the consumption of dark chocolate and cocoa products can be potentially useful in cardiovascular disorders such as CHD [150,152], stroke [150,152], atherosclerosis, HF [153], and myocardial infarction [151,154], among others. For instance, a meta-analysis of 14 prospective studies by Ren et al. [155] found that the consumption of <100 g/week of chocolate reduces the risk of having CVD (relative risk (RR) HF = 0.995, 95% CI: 0.981–1.010; RR total stroke = 0.956, 95% CI: 0.932–0.980; RR cerebral infarction = 0.952, 95% CI: 0.97–0.988; RR myocardial infarction = 0.981, 95% CI: 0.964–0.997; RR CHD = 0.986, 95% CI: 0.973–0.999). Indeed, the results showed that the most pertinent dose of chocolate consumption for reducing the risk of having CVD was 45 g/week (RR = 0.890, 95% CI: 0.849–0.932). Nonetheless, it must be taken into account that chocolate intake of >100 g/week may retract the health benefits related to chocolate by inducing adverse effects associated with high sugar consumption [155].

In the same way, a meta-analysis of a prospective study by Yuan et al. [152] concluded that moderate consumption of chocolate (1–6 servings/week) reduced the risk of developing diseases such as CHD (pooled RR = 0.90, 95% CI: 0.82–0.97), stroke (pooled RR = 0.84, 95% CI: 0.78–0.90), and diabetes (pooled RR = 0.82, 95% CI: 0.70 –0.96). Moderate consumption of chocolate has also been associated with a decreased risk of HF as seen in a recent meta-analysis of a prospective study by Gong et al. [153], although more studies are needed to determine whether this association differs according to chocolate subtypes. Likewise, in a prospective study and meta-analysis Larsson et al. [154] found that moderate chocolate consumption was related to a lower risk of myocardial infarction and ischemic heart disease.

Moreover, cocoa and dark chocolate has shown to have cardiometabolic benefits on the main risk factors related to these pathologies due to their phytochemical compounds. With regard to cocoa flavanols, a systematic review and meta-analysis of 19 RCTs was performed by Lin et al. [156] to determine the effect of this phytochemical intake on cardiometabolic biomarkers. With cocoa flavanol intake ranging from 166 to 2110 mg/day and an intervention ranging from 2 to 52 weeks, Lin et al. [156] found favorable effects with cocoa flavanol consumption on some cardiometabolic biomarkers in adults. In particular, the results showed an improvement in insulin sensitivity (WMD fasting insulin = −2.33 μIU/mL (95% CI: −3.47, −1.19 μIU/mL, *p* < 0.001), WMD insulin sensitivity index = 2.54 (95% CI: 0.63, 4.44, *p* = 0.01)), the lipid profile (WMD total TG = −0.10 mmol/L (95% CI: −0.16, −0.04 mmol/L, *p* = 0.001), and WMD HDL cholesterol = 0.06 mmol/L (95% CI: 0.02–0.09 mmol/L, *p* = 0.001)) [156].

In the case of the lipid profile, a recent systematic review and meta-analysis by González-Sarrías et al. [157] found that cocoa flavanols exhibited a significant effect on total cholesterol (*p* = −0.018), LDL-c (*p* = −0.013) and TGs (*p* = −0.047) levels. In the same way, a randomized, controlled, cross-over, free-living study carried out in healthy 24 moderately hypercholesterolemic (>2000 mg/L) subjects was performed to determine the effects of consuming 30 g of cocoa powder (containing 13.9 mg/g of soluble polyphenols) in skim milk versus milk alone during 4 weeks on the cardiometabolic profile [158]. Consumption of cocoa powder with milk improved cardiovascular health by increasing HDL-c (*p* < 0.001) levels and reducing glucose (*p* = 0.029) levels, thereby inducing hypoglycemic effects in healthy and hypercholesterolemic individuals [158]. A similar study was carried out by Martínez-López et al. [159] in 44 subjects with a mean age of 25.75 years. In this non-randomized, controlled, cross-over study of free-living individuals, the subjects consumed 200 mL of semi-skimmed milk twice a day after a two week run-in stage period. Subsequently, the individuals consumed two sachets of soluble cocoa powder/day in 200 mL of semi-skimmed milk for four weeks. The results showed that regular consumption of a flavanol-rich soluble cocoa powder in milk improves HDL-c levels in normocholesterolemic and hypercholesterolemic subjects. In contrast, a RCT by Jacobs et al. [160] studied the effect of theobromine consumption on TG and cholesterol in various lipoprotein subclass concentrations in 44 apparently healthy women and men (age: 60 ± 6 years, BMI: 29 ± 3 kg/m^2^) and found that theobromine did not have a significant effect on HDL-c in individuals with a lipoprotein phenotype characterized by low HDL-c and high TG in VLDL [160].

In relation to blood pressure, a systematic review by Ried et al. [161] showed that the consumption of flavanol-rich dark chocolate and cocoa products led to a small, albeit statistically significant, lowering of 1.8 mmHg in both SBP and DBP in mainly healthy adults in the short term.

On the other hand, the benefits of cocoa and dark chocolate consumption in CVD risk have also been associated with improvements in endothelial function [162]. West et al. [162] conducted a randomized, placebo-controlled, four-week, cross-over study in 30 middle-aged overweight adults in order to quantify the effects of dark chocolate and cocoa consumption on endothelial function and arterial stiffness. During the cocoa/chocolate intervention, the participants consumed 37 g/day of dark chocolate and a sugar-free cocoa beverage, with a total consumption of flavanols/day of about 814 mg [162]. The control group consumed a low-flavanol chocolate bar and a cocoa-free beverage mix with no added sugar, with a total consumption of flavanols/day of about 3 mg [162]. The results showed that in the group treated with cocoa, the basal diameter and peak diameter of the brachial artery significantly increased by 6% (+2 mm) and the basal blood flow volume by 22% [162], suggesting that high-flavanol cocoa and dark chocolate can enhance vasodilatation and reduce arterial stiffness. 

In relation to cardiovascular function and platelet aggregation, Rull et al. [163] compared the effects of a high-flavanol dark chocolate with a low-flavanol dark chocolate on different parameters such as blood pressure, heart rate, platelet aggregation, among others, in men with pre-hypertension or mild hypertension. They observed that the treatment with the high-flavanol dark chocolate produced a modest amelioration on cardiovascular function [163]. It should be noted that platelet aggregation seemed to be modulated by a flavanol-independent mechanism that may be due to theobromine present on cocoa or cocoa products [163]. Similar results were obtained by Pereira et al. [164] in a recent RCT evaluating the beneficial effects of long-term dark chocolate intake in 30 healthy participants from 18 to 27 years of age. It was found that regular consumption (20 g/day) of higher cocoa content chocolate provides improvement in the vascular function by reducing central brachial artery pressure and promoting vascular relaxation in healthy young adults [164].

In addition, an observational study by Montagnana et al. [165] was aimed at analyzing the effect of dark chocolate on platelet function in 18 healthy male volunteers and showed how dark chocolate consumption could be beneficial in subjects at high risk of thrombosis.

Moreover, a prospective study also examined the relationship between usual chocolate consumption and the risk of future CV events [166]. The data used to conduct this study was obtained from the European Prospective Investigation into Cancer (EPIC)-Norfolk cohort [166]. The results showed that higher chocolate consumption was related to a lower risk of future CV events (HR total CVD = 0.89, 95% CI: 0.79–1.00); HR CVD mortality = 0.75, 95% CI: 0.62–0.92) [166].

Another parameter that must undoubtedly be taken into account is inflammation, due to its clear relationship with atherosclerosis. In a systematic review and meta-analysis of RCTs, Lin et al. [156], found that cocoa flavanols intake had a beneficial effect on the inflammatory profile. The consumption of cocoa and cocoa products improved CRP (WMD = −0.83, 95% CI: −0.88, −0.77 mg/dL, *p* < 0.01) and VCAM-1 (WMD = 85.6 ng/mL, 95% CI: 16.0, 155 ng/mL, *p* = 0.02), inflammatory molecules directly related to atherosclerosis and other cardiovascular disorders. Conversely, another systematic review by Peluso et al. [167] found no clear evidence about the role of flavonoids in the modulation of the human immune system, making it necessary to obtain more evidence in humans in order to clarify this issue.

On the other hand, a RCT by Sarriá et al. [158] also found that an intervention with cocoa powder with skim milk enhanced anti-inflammatory effects by reducing IL-1 β (*p* = 0.001) and IL-10 (*p* = 0.001) levels in healthy and hypercholesterolemic individuals (Table 6).

It is known that not only inflammatory processes but also oxidative stress initiates the first stages of CVD [164]. Thus, a systematic review of 19 studies by Suen et al. [168] analyzed the role of flavonoids in oxidative stress and inflammation in adults at risk of CVD and showed that cocoa polyphenols had a great potential to reduce oxidative stress and inflammation.

Cocoa, a polyphenol-rich food, can improve CV risk factors (such as blood pressure, the lipid profile, arterial stiffness, endothelial function, oxidative stress, etc.) and prevent the negative outcome of these diseases. However, while there is a great deal of evidence regarding the beneficial effects on CVD risk, there are also a few studies that have not found the same results [160,167]. The length of the intervention, the source of the cocoa, the bioactive compound studied (theobromine, proanthocyanidins, etc.), the dose applied, or the characteristics of the subjects included may interfere with the results obtained. Further large, long-term studies are needed to clarify the beneficial effects of cocoa intake.

### 3.7. Grapes

Grapes (*Vitis vinifera*) are a fruit commonly found in the Mediterranean diet and are also one of the most popularized and extensively cultivated fruits worldwide [169]. It is calculated that about 60 species of grapes can be found today [169]. 

The phytochemicals in grapes are concentrated mainly in the skin, seeds, and juice [169]. Although resveratrol is the most known polyphenol detected in grapes, other phytochemicals such as carotenoids, flavonoids, etc. [169] can also exert important health effects.

In the last years, resveratrol (3,4’,5-trihydroxystilbene), which is found in grapes and wine [166], and pterostilbene are gaining importance as having a potentially important component on health, especially on cardiovascular health [170]. The popularity of resveratrol comes from the “French Paradox” concept, which originated in the 1980s by French epidemiologists, when low CHD death rates were found despite high dietary cholesterol and saturated fat intake [171,172]. The protective cardiovascular effects were mainly attributed to relatively high wine consumption [171,172].

Many epidemiological studies have recently suggested that grapes and their phytochemical compounds may exert a beneficial role on cardiovascular disorders by acting on its main risk factors. For instance, in the case of blood pressure, a systematic review of five studies with a total of 229 hypertensive patients by Mashhadi et al. [173] found that resveratrol seems to have anti-hypertensive effects due to certain mechanisms that allow increasing NO levels. Conversely, a recent systematic review of 39 studies published by Woerdeman et al. [174] assessed the effects of grape polyphenols on insulin sensitivity, glycaemia, blood pressure, and lipid levels and found no relevant evidence regarding a positive role of grape polyphenols on glycaemia, blood pressure, or lipid levels in individuals with or without the MetS [174].

In their meta-analyses of randomized controlled trials evaluating the effects of resveratrol on SBP and DBP, Li et al. [175] and Liu et al. [176] found that higher doses of resveratrol consumption significantly decreased SBP levels in humans, while no significant effects were found on DBP levels. Li et al. [175] reported that daily grape polyphenol intake reduced SBP by 1.48 mmHg when compared to control subjects. On the other hand, Liu et al. [176] indicated that ≥150 mg/day of resveratrol significantly reduced SBP by −11.90 mmHg (*p* = 0.01).

Moreover, in a recent RCT conducted by Neto et al. [177], 26 subjects with hypertension aged 40 to 59 years were treated with a daily dose of whole red grape juice (150 mL for men and 100 mL for women) or a control drink (control group). The individuals also performed two sessions of aerobic exercise on a treadmill (60 min, 60–85% maximum heart rate), separated by a 28-day period of the previously mentioned supplementation. At the end of the study, it was found that, in individuals with hypertension, the juice promoted a reduction in blood pressure at rest and improved post-exercise hypotension (PEH), according to the initial blood pressure values (Table 7).

In reference to wine, the effect on blood pressure of two grape extracts was tested for four weeks in a double-blind, placebo-controlled, crossover study in 60 mildly hypertensive subjects [178]. The intervention suggested that the polyphenol-rich grape-wine extract lowers ambulatory SBP and DBP; an effect that may be stronger during the daytime due to a higher blood pressure [166]. This protective effect could be explained by some mechanisms of action decreasing plasma endothelin-1 concentrations as opposed to NO, which remained unaltered [178]. Likewise, Urquiaga et al. [179] examined the effect of red wine grape pomace flour (WGPF) prepared from red wine grapes (Cabernet Sauvignon variety) on some components of the MetS in a 16-week longitudinal intervention study in 38 males, 30–65 years of age, with a minimum of one component of the MetS. At the end of the intervention, the intake of WGPF-rich in fiber and polyphenol antioxidants as a food supplement in the regular diet showed improvements in blood pressure (SBP and DBP) (*p* < 0.05), fasting glucose (*p* < 0.05), and postprandial insulin (*p* < 0.05). Additionally, antioxidant defenses increased and oxidative protein damage decreased, pointing to attenuation of oxidative stress [179].

Glucose levels have also been the object of study related to cardiovascular risk factors. For instance, in a systematic review and meta-analysis of nine articles conducted by Zhu et al. [180] resveratrol supplementation seemed to exert significant beneficial effects on T2DM by improving fasting plasma glucose (−0.29 mmol/L; 95% CI: −0.51, −0.06; *p* < 0.01), HOMA-IR (−0.52; 95% CI: −1.00, −0.04; *p* < 0.0001) and insulin as compared with placebo/control in diabetic patients. On the other hand, HbA1c levels differed, albeit not significantly, between the two groups (*−* 0.04; 95% CI: −0.48, 0.39; *p* = 0.13) [180]. Similarly, a meta-analysis of 11 studies carried out by Liu et al. [181] including a total of 388 subjects concluded that resveratrol improves glucose control and insulin sensitivity in individuals with diabetes but does not have this effect in nondiabetic subjects.

At the same time, vascular function and arterial stiffness had also been ameliorated with resveratrol treatment [182,183]. In a randomized, double-blind crossover study Pollack et al. [182], treated 30 older glucose-intolerant adults with 2–3 g/daily of resveratrol or placebo for six weeks. Resveratrol exerted beneficial effects on vascular function but not in glucose metabolism or insulin sensitivity in older adults with impaired glucose regulation. On the other hand, in a double-blind, randomized, placebo-controlled study in 50 patients with T2DM Imamura et al. [183] found that a resveratrol intervention of 100 mg/day decreased SBP (−5.5 ± 13.0 mmHg) and slightly decreased body weight (−0.8 ± 2.1 kg; *p* = 0.083) and the BMI (−0.5 ± 0.8 kg/m^2^; *p* = 0.092) compared to baseline data. These results demonstrate that resveratrol treatment may improve arterial stiffness and reduce oxidative stress as well as vascular function.

Furthermore, grapes and resveratrol consumption has also been related to having a protective role in inflammation. A recent systematic review and meta-analysis of 15 RCTs including 658 adults aged 18–75 years was performed by Haghighatdoost et al. [184]. Their main objective was to evaluate whether resveratrol supplementation could change inflammatory parameters. They found that resveratrol significantly reduced serum CRP levels (WMD = −0.54; 95% CI: −0.78, −0.30; I^2^ = 77.7%; *p* < 0.0001), but no significant effect was found in serum IL-6 (WMD = -0.06; 95% CI: -0.27, 0.14; I^2^ = 62.0%; *p* = 0.005) and TNF-α concentrations (WMD = −0.20; 95% CI: −0.55, 0.16; I^2^ = 87.2%; *p* < 0.0001) [168]. Nevertheless, resveratrol intake reduced TNF-α levels in young subjects (WMD = −0.34; 95% CI: −0.57, −0.12; I^2^ = 60.5%; *p* = 0.038) and obese individuals (WMD = −1.52; 95% CI: −2.87, −0.16; I^2^ = 74.1%; *p* = 0.004) [184]. Koushki et al. [185] obtained valuable results in their systematic review and meta-analysis of 17RCT involving 736 subjects assessing the effect of resveratrol supplementation on the levels of markers of inflammation. They observed significant reductions in TNF-α levels (WMD = −0.44; 95% CI: −0.71, −0.164; *p* = 0.002) and high-sensitivity-CRP (hs-CRP) (WMD = −0.27; 95% CI: −0.5, −0.02; *p* = 0.033), while no significant effects were seen in IL-6 levels (WMD = −0.16; 95% CI: −0.53, 0.20; *p* = 0.38) [185]. Similarly, Tabrizi et al. [186] also evaluated the possible effect of resveratrol supplementation on biomarkers of inflammation in their systematic review and meta-analysis of 24 RCTs. The results obtained showed that resveratrol supplementation significantly diminished CRP levels (SMD = −0.55; 95% CI: −0.84, −0.26; *p* < 0.001; I2: 84.0) as well as TNF-α levels (SMD = −0.68; 95% CI: −1.08, −0.28; *p* = 0.001; I2: 81.3) in patients with MetS and related disorders [186]. Conversely, IL-6 (SMD = 0.05; 95% CI: −0.31, 0.41; *p* = 0.79; I2: 85.0) and superoxide dismutase (SOD) (SMD = 0.21; 95% CI: −3.16, 3.59; *p* = 0.90; I2: 97.7) concentrations remained unchanged with resveratrol supplementation treatment [186].

In contrast, Sahebkar et al. [187] found different results in their systematic review and meta-analysis of 10 RCTs. Resveratrol supplementation did not alter plasma CRP concentrations (WMD = −0.144 mg/L, 95% CI: −0.968–0.680, *p* = 0.731) among other cardiometabolic parameters including plasma total cholesterol, LDL-c, TG, glucose levels, and blood pressure, indicating that resveratrol supplementation does not offer any positive effect in terms of CV disorders [187].

The bioactive compounds of grapes may make them useful for the prevention of some chronic diseases, such as CVD disorders, in the fight against high blood pressure, high glucose levels, as well as for ameliorating vascular function and arterial stiffness. Despite the controversial data obtained in reference to inflammation and oxidative stress, resveratrol supplementation may reduce some markers of inflammation including TNF-α and hs-CRP, although no significant effect has been described for IL-6. It is therefore necessary to establish more RCTs, with well-defined doses of resveratrol extract to obtain more in depth knowledge of the possible mechanisms by which these bioactive compounds exert their maximum effectiveness.

### 3.8. Olives

Olives, the fruit of the olive tree (*Olea europaea L.*), are botanically a drupe or stone fruit [188]. Different varieties of olives can be found on almost all the continents. In the Mediterranean region, olives are especially important as they are one of the top ingredients of Mediterranean cuisine, being the principal source of fat [189]. Natural raw olives, however, are rarely consumed because specific processes are needed to avoid the high bitterness of the fruit [190]. Additionally, several processes can be used to extract oil from olives, with mechanical processes being better than chemical processes [188]. 

Apart from the nutrients presented in these fruits, the composition of olives and virgin olive oil is also known to contain many non-nutrient compounds such as phytochemicals that have been attributed to provide a protective role against the most important risk factors to prevent and reduce the severity of the most prevalent chronic diseases, such as CV disorders [189,190,191]. The most relevant bioactive compounds found in olive oil are phenolic compounds (hydroxytyrosol and oleuropein), benzoic and cinnamic acids, polyphenols, flavonoids, and secoiridoids [190,191].

As mentioned previously, some of the risk factors typically identified for cardiovascular disorders are ameliorated by the action of olives or virgin olive oil phytochemicals. This is the case of polyphenols, a bioactive compound found in olive oil. The recent systematic review and meta-analysis by George et al. [192] showed strong evidence about the cardioprotective properties of high-polyphenol EVOO (compared to low polyphenol olive oil). High-polyphenol EVOO, improved MDA levels (*p* = 0.004), a marker of oxidative stress; as well as some lipid profile measures such as oxLDL (*p* = 0.01), total cholesterol (*p* < 0.0001) and HDL-c (*p* = 0.02). Similar results were also obtained by Schwingshackl et al. [193] in their last systematic review and network meta-analysis when comparing the impact on CV risk factors of different types of olive oil. In comparison with low-phenolic EVOO, high-phenolic EVOO provided amelioration of LDL-c (MD = −0.14 mmol/L, 95% CI: −0.28, 0.01) [193]. Additionally, it was of note that EVOO also reduced oxidized LDL-c, compared to refined olive oil [193]. The blood lipid profile was recently analyzed by Tsartsou et al. [194] in their network meta-analysis of 30 studies. The results obtained showed that adherence to the Mediterranean diet in which EVOO is the main fat used, significantly reduced total cholesterol (SD = −0.191, 95% CI: −0.259, −0.122), LDL-c (SD = −0.189, 95% CI: −0.238, −0.140), and oxidized-LDL-c (SD = −0.112, 95% CI: −0.375, 0.150), as compared to a westernized control diet (WeDiet). Furthermore, high-polyphenol olive oil significantly increased HDL-c concentrations by almost 50% (SD = 0.163, 95% CI: 0.080–0.255) [194].

On comparing different plant oils, both Ghobadi et al. [195] and Schwingshackl et al. [196] reached interesting conclusions. First, after analyzing 27 clinical trials including 1089 subjects on the effect olive oil among other plant oils, on the blood lipid profile, Ghobadi [195] found significantly relevant reductions in total cholesterol (6.27 mg/dL, 95% CI: 2.8–10.6), LDL-c (4.2 mg/dL, 95% CI: 1.4–7.01), and TG (4.31 mg/dL, 95% CI: 0.5–8.12) with olive oil. Second, the systematic review and network meta-analysis by Schwingshackl et al. [196] observed that vegetal unsaturated fatty rich oils (olive, sunflower, safflower, corn, etc.) were more effective in reducing lipid profile measures such as LDL-c, as compared with saturated fatty acid–rich animal products, like butter or lard, used in many cultures. Similar to the study by Schwingshackl et al. [196], Khaw et al. [197] carried out a randomized trial aimed at quantifying the effect of coconut oil, olive oil or butter on blood lipids in volunteer men and women aged 50–75 years with no known history of CVD, diabetes, or cancer (Table 8). The study participants were randomized to receive 50 g/day for four weeks of extra virgin coconut oil (EVCO), EVOO or unsalted butter group [197]. In the butter group, LDL-c concentrations were significantly increased compared with EVCO (0.42, 95% CI: 0.19–0.65 mmol/L, *p* < 0.0001) and EVOO (0.38, 95% CI: 0.16–0.60 mmol/L, *p* < 0.0001), while no relevant differences were observed in LDL-c change with EVCO or EVOO (−0.04, 95% CI: −0.27 to 0.19 mmol/L, *p* = 0.74) [185]. HDL-c levels were significantly increased with EVCO in comparison with butter (0.18, 95% CI: 0.06–0.30 mmol/L) or EVOO (0.16, 95% CI: 0.03–0.28 mmol/L) [197]. Moreover, the total cholesterol/HDL-c ratio and non-HDL-c also increased in the butter subgroup compared with the EVCO group [197].

In addition, a randomized, double-blind, controlled, crossover trial by Lockyer et al. [198] was aimed at examining the effects of a phenolic-rich olive leaf extract (OLE) on different metabolic parameters of 60 pre-hypertensive individuals and found lipid-lowering effects following OLE intake. Reductions in total plasma cholesterol (*p* = 0.002), LDL-c (*p* = 0.017) and TG (*p* = 0.008) were observed in comparison with the control group [198].

On the other hand, with regard to blood pressure, a reduction in SBP was observed using both high- and low-phenolic EVOO compared to refined olive oil (range of MD = −2.99 to −2.87 mmHg) [193]. The intake of 10–50 mL/day of EVOO also showed a decrease in blood pressure, especially DBP (*p* < 0.001), while reductions of SBP were not statistically significant in the systematic review and meta-analysis by Zamora-Zamora et al. [199]. 

Body weight has also been identified as one of the main risk factors for developing cardiovascular disorders. Popular knowledge has sometimes attributed weight-gain properties to the consumption of olive oil. However, several epidemiological studies such as the recent systematic review and meta-analysis of 11 RCTs by Zamora-Zamora et al. [200] observed that a diet high in olive oil may contribute to be an important weight control strategy in people without previous cardiovascular problems.

In relation to inflammation and endothelial function, there are many studies on the beneficial effects of olive oil or one of its phytochemicals in ameliorating these cardiovascular risk factors. In their recent systematic review and meta-analysis, Fernandes et al. [201] analyzed the effect of olive oil on some biomarkers of inflammation. They reported a strong beneficial anti-inflammatory effect on reducing IL-6 levels (*p* < 0.001), while no significant changes were observed for CRP levels (*p* = 0.61) [201]. Conversely, in their previous systematic review and meta-analysis of 30 studies with a total of 3106 participants, Schwingshackl et al. [202] found that olive oil treatment decreased CRP (MD = −0.64 mg/L; 95% CI: −0.96 to −0.31, *p* < 0.0001, I^2^ = 66%) concentrations compared to their respective controls. Moreover, IL-6 levels also decreased with olive oil interventions (MD = −0.29; 95% CI: −0.7 to −0.02, *p* < 0.04, I^2^ = 62%) compared to controls.

On the other hand, in a study on the effect of olive oil flavonoids on oxidative stress in adults at high risk of CVD, Suen et al. [168] observed the great potential of these flavonoids to reduce oxidative stress and inflammation in hypercholesterolemic and hypertensive subjects.

Finally, although the other markers of inflammation and vascular function studied remained unchanged, in the RCT by Lockyer et al. [198], reductions in IL-8 (*p* = 0.026) were noted with an intervention with OLE compared to controls.

Thus, due to the phytochemicals, such as polyphenols in olives and EVOO, their consumption can provide some beneficial effects such as a reduction of blood pressure, a lipid-lowering function, an improvement of endothelial function and better control of body weight. We should remember that a daily intake of 5.6 mg of hydroxytyrosol has positive benefits for our health [27]. In relation to oxidative stress and inflammatory biomarkers, although the results among studies differ, an amelioration of these markers should be noted. Additionally, more studies are required, especially in non-Mediterranean cultures where the consumption of olives and EVOO is lower than in Mediterranean countries.

## 4. Conclusions

According to the evidence currently available, various fruit and vegetable bio-compounds (flavonoids, sulfur, quercetin, allium, stilbene, etc.) seem to exert numerous benefits against atherosclerosis, whether consumed as whole fruits and vegetables and their bioactive compounds or as supplements. Nevertheless, while many factors might explain the differences between the studies that failed to demonstrate the possible benefits of the direct action of the bioactive compound of a food consumed, the usual dose of these foods may be very different from the dose actually needed to achieve any beneficial effect.

Although there is currently a large amount of scientific evidence regarding the benefits of F/V against chronic inflammatory diseases such as T2DM, cancer, or CVD, the highest level of evidence should be based on the results of well-designed, large, long-term RCTs.

Therefore, before F/V can be proposed as a tool to aid in the prevention of NCDs such as MetS, obesity, cognitive disorders, or the risk of mortality, more clinical trials are needed to confirm their potential beneficial effects.

## Figures and Tables

**Figure 1 nutrients-11-02381-f001:**
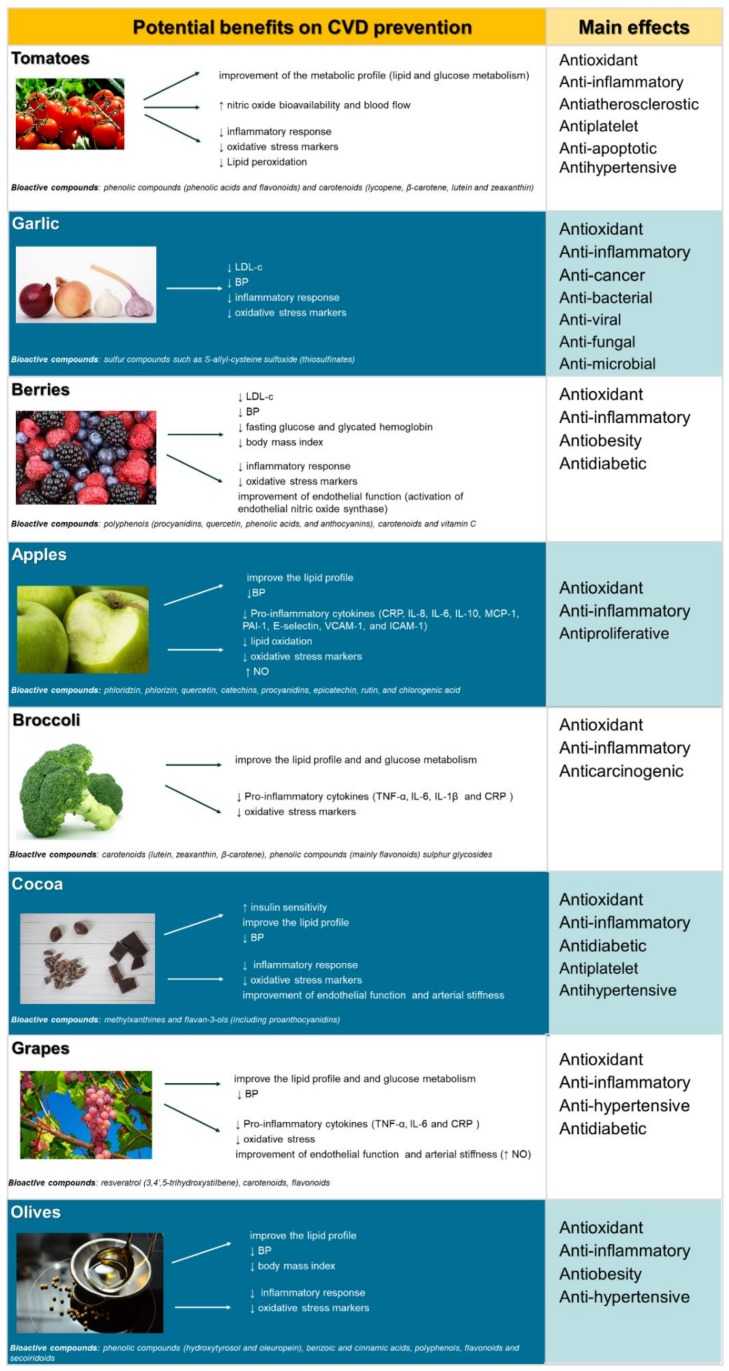
Summary of the potential health benefits of the main bioactive compounds contained in fruits and vegetables mentioned in this study.

**Table 1 nutrients-11-02381-t001:** Potential protective effect of tomatoes or their bioactive compounds on proinflammatory and oxidative stress markers during the progression of atherosclerosis.

Reference	Study Design	Study Duration	Participants	Type of Study	Findings Plasma Lipids Analyzed Markers	
Valderas-Martinez et al. [53]	A single dose of 7.0 g of RT/kg of BW, 3.5 g of TS/kg BW, 3.5 g of TSOO/kg BW and 0.25 g of sugar dissolved in water/kg BW on a single occasion on four different days.	Acute	40 healthy subjects	Open, prospective, randomized, cross-over, controlled feeding trial.	↓ TC, TG and ↑ HDL.	↑ IL-10 and ↓ MCP-1 TSOO: ↓ VCAM-1, IL-6, LFA-1 (lymphocytes) and CD36 (monocytes).
Burton-Freeman et al. [55]	Consumed high-fat meals of processed tomato product or non-tomato on two separate occasions.	Acute	25 participants (mean age = 27 ± 8 years; mean BMI= 22 ± 2).	Single center, randomized, cross-over, two-arm, two-sequence, placebo-controlled, 360-min postprandial trial.		↓OxLDL (*p* < 0.05) and ↑ IL-6 (*p* < 0.0001).
Li et al. [50]	280 mL of tomato juice (32.5 mg of lycopene) daily for 8-weeks.	Short	30 young females, aged between 20 to 30 years with a BMI ≥20.	Uncontrolled supplementation trial	↓ waist circumference and cholesterol levels.	↓ MCP-1, ↑ adiponectin.
García-Alonso et al. [51]	500 mL of *n*-3 PUFA-enriched juice (181 mg of phenols and 26.5 mg of lycopene) daily for 2-weeks.	Short	18 healthy women (aged 35–55 years) and BMI: 21–30.	Randomized single-blind intervention trial.		↓ ICAM-1, VCAM-1 and homocysteine ↓ MDA ↑ antioxidant capacity ↑β-Carotene and vitamin C.
Deplanque et al. [56]	Lycopene and phytosterols in a 1:1 ratio (15 mg) or placebo for 2-weeks.	Short	146 healthy normal weight individuals (BMI: ≥18.5 and <25), aged 18–70 year.	Randomized, double-blind, parallel-groups, placebo-controlled study.		↓OxLDL (*p* < 0.0001) No changes in glucose, insulin, or TG levels.
Ghavipour et al. [57]	330 mL of tomato juice (37.0 mg of lycopene) or water daily for 20 days.	Short	64 overweight or obese (BMI ≥ 25) female students	Randomized controlled clinical trial.		↑ TAC ↑ SOD, GPx and CAT. ↓ MDA
Pourahmadi et al. [58]	330 mL of tomato juice (60 mg of lycopene) or water daily for 20 days.	Short	75 overweight or obese female students, aged 20 to 30 years, and BMI ≥ 25.	Randomized controlled clinical trial.		No changes in SOD, GPx or CAT.
Thies et al. [54]	<10 mg lycopene /day; 32–50 mg lycopene/day; 10 mg lycopene /day for 3-months.	Short	225 volunteers aged 40–65 y and BMI: 18.5 and 35.	Single-blind, randomized controlled intervention trial.	No changes in markers of insulin resistance or sensitivity.	No changes in inflammatory markers.

BMI, Body mass index; BW, Body weight; CAT, Catalase; GPx, Glutathione peroxidase; HDL, High-density lipoprotein; ICAM-1, Intercellular adhesion molecule-1; IL-, Interleukin; LDL, Low-density lipoprotein; MCP-1, Monocyte chemoattractant protein-1; MDA, Malondialdehyde; OxLDL, Oxidized low-density lipoprotein; PUFA, Polyunsaturated fatty acids; RT, Raw tomatoes; SOD, Superoxide dismutase; TC, Total cholesterol; TG, Triglyceride; TS, Tomato sauce; TSOO, Tomato sauce with refined olive oil; VCAM-1, Vascular adhesion molecule-1.

**Table 2 nutrients-11-02381-t002:** Potential protective effect of garlic or its bioactive compounds on proinflammatory and oxidative stress markers during the progression of atherosclerosis.

Reference	Study Design	Study Duration	Participants	Type of Study	Findings Plasma Lipids Analyzed markers	
Wang et al. [92]	Daily doses of 9 g Black Garlic or placebo during 14 days.	Short	19 healthy, nonsmoking and untrained males (22.8 ± 6.0).	Double-blind, parallel design study.		↓dROMs, lipid peroxide, 8-iso-prostaglandin F2α.
Atkin et al. [89]	1200 mg of AGE or placebo daily for four weeks.		26 subjects with T2DM, aged 18 to 70 years.	Double blind, placebo-controlled crossover pilot study.		No changes in TAOS, GSH/GSSG, LHP or CRP, IL-6.
Williams et al. [90]	2.4 g/day of AGE or placebo for two weeks.	Short	15 men with angiographically proven CAD aged 45 to 70 years.	Randomized, placebo-controlled, cross-over design.		No changes in oxLDL and peroxides, CRP and IL-6 and endothelial activation (VCAM-1) ↑ FMD.
Zare et al. [86]	400 mg of standardized garlic extract twice a day or placebo for two months.	Short	42 peritoneal dialysis patients, aged 18 to 80 years.	Parallel-designed double blind randomized clinical trial.		↓ IL-6 and CRP.
Kumar et al. [87]	Control group and 500 mg/day garlic extract of Allium sativum twice a day for 12 weeks.	Short	60 patients with T2DM and obesity.	Open-label, prospective, comparative study.	↓Fasting blood glucose and postprandial blood glucose. ↓ TC, LDL and TG and ↑ HDL	↓ CRP and adenosine deaminase.
Ried et al. [70]	Daily intake of 1.2 g of AGE (1.2 mg S-allylcysteine) or placebo for 12 weeks.	Short	49 participants with uncontrolled hypertension (SBP ≥ 140 mmHg and/or DBP ≥ 90 mmHg).	Double-blind randomized placebo-controlled trial.	↓ SBP (10 ± 3.6 mmHg) and DBP (5.4 ± 2.3 mmHg). ↓ Pulse pressure and arterial stiffness (*p* < 0.05).	↓ TNF-α and IL-6 Improvement of gut microbiota (↑ *Lactobacillus* and *Clostridia* species).
Zeb et al. [88]	Daily intake of AGE (1200 mg) plus CoQ10 (120 mg) or placebo for one year.	Intermediate	65 intermediate risk (CAC score >10 at baseline) firefighters (mean age: 55 ± 6 years).	Placebo-controlled, double-blind, randomized trial.		AGE+CoQ10: ↓ CAC progression (32 ± 6 vs. 58 ± 8, *p* = 0.01) and ↓ CRP (−0.12 ± 0.24 vs. 0.91 ± 0.56 mg/L, *p* < 0.05).
Liu et al. [91]	20 g garlic (2.60 mg GAeq/g) daily or placebo for six months.	Intermediate	120 chronic heart failure patients caused by CHD. Age 35–75 years.	Randomized controlled clinical trial.		↓ Nt-proBNP ↑ Circulating antioxidant levels.

AGE, Aged garlic extract; CAC, Coronary artery calcium; CAD, Coronary artery disease; CoQ10, Coenzyme Q10; CHD, Coronary heart disease; CRP, C-reactive protein; dROMS, Reactive oxygen metabolites; DBP, Diastolic Blood Pressure; FMD, Flow-mediated dilation; GAeq, Eq gallic acid; GSH/GSSG, Reduced and oxidized glutathione; HDL, High-density lipoprotein; IL-, Interleukin; LDL, Low-density lipoprotein; LPH, Plasma lipid hydroperoxides; Nt-proBNP, BNP precursor N-terminal; OxLDL, Oxidized low-density lipoprotein; SBP, Systolic blood pressure; T2DM, Type 2 diabetes mellitus; TC, Total cholesterol; TAOS, Plasma total antioxidant status; TNF-α, Tumor necrosis factor-α; TG, Triglyceride; VCAM-1, Vascular adhesion molecule-1.

**Table 3 nutrients-11-02381-t003:** Potential protective effect of berries or their bioactive compounds on proinflammatory and oxidative stress markers during the progression of atherosclerosis.

Reference	Study Design	Study Duration	Participants	Type of Study	Findings Plasma Lipids Analyzed Markers	
Lee et al. [106]	Daily intake of 2.5 g of anthocyanin-rich (12.58 mg/g extract) black soybean test extracts or placebo for two months.	Short	63 overweight or obese participants, with BMI > 23 or waist circumference > 90 cm for males, >85 cm for females. Age 19 to 65 years.	Randomized, double-blinded, and placebo-controlled clinical trial.	↓ abdominal fat, TG and LDL ↓ TC/HDL and LDL/HDL	↓ MCP-1 (*p* = 0.031) and TNF-α (*p* = 0.011)
Soltani et al. [108]	Twice daily intake of 500 mg of dried granules equivalent to 45 ± 2 mg of total anthocyanin or placebo for one month.	Short	50 hyperlipidemic adult patients (age ≥ 18 years).	Randomized, double-blind, placebo-controlled clinical trial.	↓ TC (*p* < 0.001), LDL (*p* = 0.004), TG (*p* < 0.001). No changes in HDL	MDA (*p* = 0.013) No changes in CRP
Alvarez-Suarez et al. [109]	Intake of 500 g of strawberries for one month.	Short	23 healthy volunteers (age 27 ± 3.2 and BMI 21.74 ± 2.5 kg/m^2^).	Randomized, double-blind, placebo-controlled clinical trial.	↓ TC (−8.78%), LDL (−13.72%) and TG levels (−20.80%).	↓ MDA, urinary 8-OHdG and isoprostane levels (*p* < 0.05; all)
Davinielli et al. [110]	Three capsules of 150 mg standardized maqui berry extract containing 54 mg of anthocyanins daily or placebo for four weeks.	Short	42 healthy participants, aged 45–65 years and BMI between 25 and 30 kg/m^2^.	Randomized, double-blind, placebo-controlled clinical trial.		↓ oxLDL and 8-iso-prostaglandin F2α
Yang et al. [111]	Daily intake of 320 mg of purified anthocyanins (from bilberry and blackcurrant) or placebo for 12 weeks.	Short	138 volunteers aged 40−75 years with prediabetes or early untreated diabetes	Randomized, double-blind, placebo-controlled clinical trial.	↓ HbA1c (−0.14%, *p* = 0.005), LDL (−0.2 mmol/L, *p* = 0.04), apoA-1 (0.09 g/L, *p* = 0.02), and apo B (−0.07 g/L, *p* = 0.01)	No changes in CRP levels.
Zhang et al. [107]	Daily intake of 320 mg of purified anthocyanins or placebo for six months.	Intermediate	146 hypercholesterolemic individuals. Age from 40 to 65 years	Randomized, double-blind, placebo-controlled trial.		↓ CXCL7 (−12.32% vs. 4.22%, *p* = 0.001), CXCL5 (−9.95% vs. 1.93%, *p* = 0.011), CXCL8 (−6.07% vs. 0.66%, *p* = 0.004), CXCL12 (−8.11% vs. 5.43%, *p* = 0.023) and CCL2 levels (−11.63% vs. 12.84%, *p* = 0.001). ↓hs-CRP, IL-1β and sP-selectin

apoA-1, Apolipoprotein A1; ApoB, Apolipoprotein B; BMI, Body mass index; CXCL, Chemokine; HbA1c, Glycosylated hemoglobin; HDL, High-density lipoprotein; hs-CRP, high sensitivity C-reactive protein; IL-, Interleukin; LDL, Low-density lipoprotein; MCP-1, Monocyte chemoattractant protein-1; MDA, Malondialdehyde; OxLDL, Oxidized low-density lipoprotein; TC, Total cholesterol; TNF-α, Tumor necrosis factor-α; TG, Triglyceride; 8-OHdG, 8-hydroxy-2′-deoxyguanosine.

**Table 4 nutrients-11-02381-t004:** Potential protective effect of apples or their bioactive compounds on proinflammatory and oxidative stress markers during the progression of atherosclerosis.

Reference	Study Design	Study Duration	Participants	Type of Study	Findings Plasma Lipids Analyzed Markers	
Bondonno et al. [124]	Four energy-matched treatments: control (low-flavonoid apple control and low-nitrate control), apple (high-flavonoid apple active and low-nitrate control), spinach (low-flavonoid apple control and nitrate-rich spinach active), and apple+spinach (high-flavonoid apple active and nitrate-rich spinach active) at 100, 150, and 200 min after lunch/intervention (acute effects).	Kinetic	30 healthy men and women.	A randomized, controlled, crossover trial.		↑ FMD ↑ nitric oxide status
Soriano-Maldonado et al. [122]	Twice daily intake of 250 mL/day of apple juice (60 mg/L vitamin C and 510 mg catechin equivalent/L) or a polyphenol-rich juice (22 mg/L vitamin C and 993 mg catechin equivalent/L) for one month.	Short	20 healthy subjects, aged 21–29 years, BMI ≤ 27.5 kg/m^2^.	A randomized cross-over trial.		↑ Plasma antioxidant activity (*p* = 0.031). ↓ IL-8, IL-6, IL-10, MCP-1, PAI-1, E-selectin, VCAM-1, and ICAM-1.
Zhao et al. [125]	Three treatments: 1) one apple per day; 2) an apple extract in capsules (twice daily, 194 mg polyphenols/day); 3) Control group for one month.	Short	51 healthy middle-aged adults (aged 40–60 years old)	Randomized, double-blind, placebo-controlled clinical trial.		Whole apple and extract: ↓oxLDL-β2GPI. No changes in SOD.
Auclair et al. [126]	Daily intake of 40 g of two lyophilized apples: 1) polyphenol-rich (1.43 g of polyphenols per day); 2) polyphenol-poor (0.21 g of polyphenols per day) for four weeks.	Short	30 hypercholesterolemic volunteers, with a mean age of 52.6 ± 5.5 years, a mean BMI of 25.7 ± 2.6.	Double-blind, randomized crossover trial.	No changes in lipid profile.	No changes in FMD, homocysteine, antioxidant capacity
Saarenhovi et al. [127]	Daily intake of apple polyphenol extract (100 mg epicatechin and flavan-3-ol) or placebo for four weeks.	Short	81 otherwise healthy participants aged 40–65 years, with borderline hypertension or unmedicated mild hypertension.	Single-center, repeated-dose, double-blind, placebo-controlled, crossover study		No significant changes were observed in NMD, CRP, E-selectin, VCAM-1 or ICAM-1, ADMA, vWF, PAI-1 and asymmetric dimethylarginine.
Chai et al. [123]	Daily intake of 75 g of dried apple or 75 g of dried plum (comparative control) for 12 months.	Intermediate	160 healthy postmenopausal women (1–10 years after menopause), without hormonal treatment.	Randomized, double-blind, placebo-controlled clinical trial.	↓ TC (9%) and LDL (16%) at 3 months ↓ TC (13%) and LDL (24%) at 6 months.	↓ lipid hydroperoxide and CRP

ADMA, Asymmetric dimethylarginine; BMI, Body mass index; CRP, C-reactive protein; FMD, Flow-mediated dilation; ICAM-1, Intercellular adhesion molecule-1; IL-, Interleukin; LDL, Low-density lipoprotein; MCP-1, Monocyte chemoattractant protein-1; NMD, Endothelium-independent nitrate-mediated vasodilatation; oxLDL-β2GPI, oxidized low-density lipoprotein/beta2-glycoprotein I complex; PAI-1, Plasminogen activator inhibitor-1; SOD, Superoxide dismutase; TC, Total cholesterol; VCAM-1, Vascular adhesion molecule-1; vWF, von Willebrand factor.

**Table 5 nutrients-11-02381-t005:** Potential protective effect of broccoli or its bioactive compounds on proinflammatory and oxidative stress markers during the progression of atherosclerosis.

Reference	Study Design	Study Duration	Participants	Type of Study	Findings Plasma Lipids Analyzed Markers	
Mirmiran et al. [143]	Daily intake of 10 g (225 µmol sulforaphane), 5 g (112 µmol sulforaphane) of broccoli sprouts powder or placebo for four weeks.	Short	81 patients with T2DM aged 18–60 years.	Parallel, randomized, double-blind, and placebo controlled clinical trial.		↓ hs-CRP (10 g: −20.5% and 5g: −16.4%). No significant changes in TNF-α or IL-6.
Bahadoran et al. [140,144]	Daily intake of 10 g (225 µmol sulforaphane), 5 g (112 µmol sulforaphane) of broccoli sprouts powder or placebo for four weeks.	Short	81 patients with T2DM, aged 18–60 years	Parallel, randomized, double-blind, and placebo controlled clinical trial.	10 g/day: ↑ HDL (*p* < 0.01 for treatment).	↓ MDA (*p* = 0.001 for treatment effect), oxLDL (*p* = 0.03 for treatment effect), OSI (*p* = 0.001 for treatment effect) and ↑ TAC (*p* = 0.001 for treatment effect). 10g/day: ↓oxLDL/LDL ratio, TG and AIP (*p* < 0.05 for treatment effect)
Lopez-Chillón et al. [145]	Daily consumption of broccoli sprouts (30 g/day) for 10 weeks. After, normal diet without broccoli sprouts intake for 10 weeks (the follow-up phase).	Short	40 no- smoking overweight subjects. Aged 35–55 years and BMI from 24.9 to 29.9 kg/m^2^	Interventional follow-up study.		↓ IL-6 significantly decreased (−38%). ↓ CRP (−59%, *p* < 0.05).
Jiang et al. [146]	Evaluate association between vegetable intake with inflammatory and oxidative stress markers.		1005 women (40–70 y of age) selected from the Shanghai Women’s Health Study (SWHS).	Cross-sectional study.		↓ TNF-α (−2.66%, *p* = 0.01), IL-1β (−18.18%, *p* = 0.02), and IL-6 (−24.68%, *p* = 0.02).

AIP, Atherogenic index of plasma; BMI, Body mass index; CRP, C-reactive protein; HDL, High-density lipoprotein; hs-CRP, high sensitivity C-reactive protein; IL-, Interleukin; MDA, Malondialdehyde; OSI, Oxidative stress index; OxLDL, Oxidized low-density lipoprotein; T2DM, Type 2 diabetes mellitus; TAC, Plasma total antioxidant capacity; TNF-α, Tumor necrosis factor-α; TG, Triglyceride.

**Table 6 nutrients-11-02381-t006:** Potential protective effect of cocoa or its bioactive compounds on proinflammatory and oxidative stress markers during the progression of atherosclerosis.

Reference	Study Design	Study Duration	Participants	Type of Study	Findings Plasma Lipids/Others Analyzed Markers	
Sarriá et al. [158]	Intervention group consuming two servings/day (15 g each) of a cocoa product rich in fiber in milk vs. control group consuming only milk for one month.	Short	Healthy (*n* = 24) and moderate hypercholesterolemic (>2000 mg/L, *n* = 20) subjects.	Randomized, controlled, cross-over, free-living study	↑ HDL-c concentration ↓fasting serum glucose levels.	↓ IL-1β and IL-10
Martínez-López et al. [159]	Intervention group consumed two servings/day (7.5 g per serving) of a soluble cocoa product in milk vs. control group only taking milk for one month.	Short	Healthy (*n* =24) and moderate hypercholesterolemic (200–240 mg/dL, *n* = 20) individuals.	Non-randomized, controlled, crossover, free-living study	↑ HDL-c and dietary fiber intake	↓ IL-10
Jacobs et al. [160]	Intervention group consumed a drink supplemented with 500 mg/day theobromine vs. control group for one month.	Short	Apparently healthy women and men with low baseline HDL-c levels.	Randomized, double-blind, placebo-controlled, cross-over study	Theobromine showed no effect on HDL-c in subjects characterized by low HDL-c and high TG in VLDL.	
West et al. [162]	Intervention group consumed 37 g/day of dark chocolate and a sugar-free cocoa beverage vs. control group for one month.	Short	Middle-aged, overweight adults.	Randomized, placebo-controlled, cross-over study		↑ The basal diameter and peak diameter of the brachial artery by 6% (+2 mm) and basal blood flow volume by 22%

HDL-c, High-density lipoprotein-cholesterol; IL-, Interleukin; TG, Triglyceride; VLDL, very low-density lipoprotein.

**Table 7 nutrients-11-02381-t007:** Potential protective effect of grapes or their bioactive compounds on proinflammatory and oxidative stress markers during the progression of atherosclerosis.

Reference	Study Design	Study Duration	Participants	Type of Study	Findings Plasma Lipids/ Others	
Neto et al. [177]	Experimental group taking whole red grape juice (*n* = 14) and control group taking a control drink (*n* = 12) for 28-day period.	Short	26 individuals with hypertension aged from 40 to 59 years old.	Double-blind, randomized controlled study	Whole red grape juice promotes reduces BP at rest and improves post-exercise hypotension in hypertensive subjects.	
Draijer et al. [178]	Placebo group, Grape juice extract alone group and Mixture of grape and wine extract group for four weeks.	Short	60 males and females mildly hypertensive aged ≥35 and ≤75 years.	Double-blind placebo-controlled crossover study	The intervention with grape juice extract alone had no effect on BP while the polyphenol-rich grape-wine extract lowered SBP and DBP.	
Urquiaga et al. [179]	Intervention group taking 20 g/day of wine grape pomace flour (*n*= 25) and control group (*n* = 13) for 16 weeks.	Short	38 males aged from 30 to 65 years old with at least one component of MetS.	Randomized controlled trial	Improvement of BP, glycemia and postprandial insulin.	
Pollack et al. [182]	Resveratrol group (2–3 g/day) and placebo group for six weeks.	Short	30 older glucose-intolerant adults aged 50-80 years old without a prior diagnosis of diabetes	Randomized, double-blind crossover study	No effects on glucose metabolism or insulin sensitivity	Beneficial effects on vascular function

BP, Blood Pressure; DBP: Diastolic Blood Pressure; MetS, Metabolic syndrome; SBP, Systolic blood pressure.

**Table 8 nutrients-11-02381-t008:** Potential protective effect of olives or their bioactive compounds on proinflammatory and oxidative stress markers during the progression of atherosclerosis.

Reference	Study Design	Study Duration	Participants	Type of Study	Findings Plasma Lipids/Others
Khaw et al. [197]	EVCO group (50 g/day, *n* = 30), EVOO group (50 g/day, *n* = 33) and unsalted butter group (50 g/day, *n* = 33) for four weeks.	Short	94 men and women aged 50–75 years, with no known history of cancer, CVD or diabetes, not on lipid lowering medication and no contra-indications to a high-fat diet.	Randomized clinical trial	EVCO and butter, may have different effects on blood lipid profile compared with EVOO with respect to LDL-c.
Lockyer et al. [198]	Liquid OLE supplement group (136 mg oleuropein; 6 mg hydroxytyrosol) and a control polyphenol-free group for six weeks.	Short	60 pre-hypertensive males aged 24–72 years.	Double-blind, randomized, controlled, crossover trial	OLE intake confers lipid-lowering and hypotensive effects.

CVD: Cardiovascular disease; EVCO, extra virgin coconut oil; EVOO, extra virgin olive oil; LDL-c, Low-density lipoprotein cholesterol; OLE, olive leaf extract.

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
