# Peer review of "Relation of Fruits and Vegetables with Major Cardiometabolic Risk Factors, Markers of Oxidation, and Inflammation"

_nutrients, 2019, doi:10.3390/nu11102381_

Round 1
Reviewer 1 Report
To:
Editorial Board
Nutrients
Title: “Key foods in the prevention of atherosclerosis”
Dear Editor,
I read this manuscript and I think that:
the authors should include a representative figure for this paper. Please provide. The authors should discuss about the role of nutraceuticals in such a setting. Please consider and discuss the paper from Scicchitano P et al. Journal of Functional Foods 2014;6:11-32.
Author Response
nutrients-581555
REVIEWER #1:
The authors should include a representative figure for this paper. Please provide.Comment 1: Thank you for your comment. We have included a representative figure.
The authors should discuss about the role of nutraceuticals in such a setting. Please consider and discuss the paper from Scicchitano P et al. Journal of Functional Foods 2014;6:11-32.Comment 2: Thank you again for your comment. We have included this reference in the Introduction section. The changes are highlighted in yellow (lines 59-68).
Lines 59-68: ”According to Scicchitano et al. [15], to date, there is vast scientific evidence about the effectiveness of nutraceuticals against the development of CVD and in reducing the burden of the atherosclerotic process. Nevertheless, how these nutraceuticals exert their positive action on the cardiovascular system is not yet known. It should be highlighted that the positive health effects of these bioactive compounds have mainly been based on observational, in vitro and in vivo studies [15]. While there is sufficient in vitro evidence supporting the antioxidant, anti-inflammatory, and antidiabetic effects, among others, of allicin, flavonoids or stilbenes, the in vivo studies available have reported controversial results, indicating that further research are required. To demonstrate and corroborate the results observed in these in vitro and in vivo studies, large controlled clinical trials are needed in humans [15-19].”

Reviewer 2 Report
General comments
The aim of this work is to review the recent evidence correlating F/V consumption and its role in the prevention of atherosclerosis and to investigate and quantify the magnitude of the beneficial effects observed and analyse the potential pathways by which selected food-phytochemicals exert their positive effects.
The topic is certainly interesting, but several aspects must be improved.
Most of the papers reviewed explore the relation of foods/nutrients with major cardiometabolic risk factors and markers of oxidation/ inflammation, very few deal with surrogate measures of atherosclerosis, such as markers of endotelial disfunction, and only one measures progression of coronary artery calcification. The potential pathways of action are insufficiently described. For these reasons I would suggest that the Authors consider rephrasing the title and the study aim to make them more coherent with the content of the review.
Specific comments
The literature on this topic is quite extensive, the Authors must clearly define the criteria for the bibliographical search and the criteria for inclusion/exclusion of the articles in the review.
Table 1 is difficult to read. The authors may consider creating a table for each food and within each table dividing studies according to end points (plasma lipids, markers of imflammation etc..) and study duration (i.e acute or short /intermediate duration). The magnitude of the changes observed for each end point might also be reported in these tables. This would help presenting the results in a more organized form and would facilitate comparative evaluation between studies.
For each food the conclusions must summarize the results relative to the different end points. A comment should be included on the magnitude of the changes observed and on whether they are clinically relevant.The possible reasons for discrepancies between studies (i.e. dosage of substance used, concomitant use of other substances which may modulate the effect, study population, study duration etc…) should be discussed and efforts should be made to identify crucial methodological problems which need to be overcome in future studies.
It would also be relevant to include a comment on how the dosage of the compound(s) used in the different studies compares with habitual intake and whether these quantities can be reached with the use of natural foods or only by use of supplemets.
Author Response
nutrients-581555
RESPONSES TO REVIEWERS:
REVIEWER #2:
The aim of this work is to review the recent evidence correlating F/V consumption and its role in the prevention of atherosclerosis and to investigate and quantify the magnitude of the beneficial effects observed and analyse the potential pathways by which selected food-phytochemicals exert their positive effects.
The topic is certainly interesting, but several aspects must be improved.
Most of the papers reviewed explore the relation of foods/nutrients with major cardiometabolic risk factors and markers of oxidation/ inflammation, very few deals with surrogate measures of atherosclerosis, such as markers of endothelial disfunction, and only one measures progression of coronary artery calcification. The potential pathways of action are insufficiently described. For these reasons I would suggest that the Authors consider rephrasing the title and the study aim to make them more coherent with the content of the review.Comment 1.1: Thank you for your comment. We have changed the title following your suggestion. The changes are highlighted in yellow (lines 2-4).
Lines 2-4: “Relation of fruits and vegetables with major cardiometabolic risk factors, markers of oxidation and inflammation”
Comment 1.2: Thank you again. Following the suggestion of Reviewer #2, we have changed the aim of the study to make it more coherent with the content of the review.The changes are highlighted in yellow (lines 91-93).
Lines 91-93: “The main objective of this review was to perform a bibliographical search to better understand the most recent evidence available correlating F/V consumption with major cardiometabolic risk factors, markers of oxidation and inflammation.”
Specific comments
The literature on this topic is quite extensive; the Authors must clearly define the criteria for the bibliographical search and the criteria for inclusion/exclusion of the articles in the review.Comment 2: Thank you for your comment.This is not a systematic review, and therefore, we do not think that this information is necessary in a Review such as ours. However, some terms have been considered and included with the changes being highlighted in yellow (84-89).
Lines 84-89: “In this article, a bibliographic review was carried out using the PubMed, Science Direct, Scopus, Cochrane Library databases It was written based on the most relevant articles and studies made in human subjects published no longer than approximately 7 years ago and reviewed in the English language literature of humans with no time restriction. The keywords used for this search were: fruits and vegetables, atherosclerosis, bioactive compounds, immune system, inflammation, inflammatory markers, oxidative stress, cytokines, phytochemicals, nutraceuticals, etc.”
Table 1 is difficult to read. The authors may consider creating a table for each food and within each table dividing studies according to end points (plasma lipids, markers of imflammation etc.) and study duration (i.e acute or short /intermediate duration). The magnitude of the changes observed for each end point might also be reported in these tables. This would help presenting the results in a more organized form and would facilitate comparative evaluation between studies.Comment 3: Thank you for your comment again. We have created new tables, one for each section (food). In addition, according to your suggestion, we have divided the studies according to end points and study duration. The changes are highlighted in yellow (Table 1-8).
For each food the conclusions must summarize the results relative to the different end points. A comment should be included on the magnitude of the changes observed and on whether they are clinically relevant.The possible reasons for discrepancies between studies (i.e. dosage of substance used, concomitant use of other substances which may modulate the effect, study population, study duration etc…) should be discussed and efforts should be made to identify crucial methodological problems which need to be overcome in future studies.Comment 4: We agree with the reviewer. However, because of the lack of time we have slightly modified the summary at the end of each food analyzed. We hope this is sufficient. The changes are highlighted in yellow (Lines 230-240; 307-314; 405-417; 491-498; 563-573; 691-698; 795-802 and 889-896).
It would also be relevant to include a comment on how the dosage of the compound(s) used in the different studies compares with habitual intake and whether these quantities can be reached with the use of natural foods or only by use of supplements.Comment 5: Thank you for your suggestion. We have introduced a new section in the manuscript in which we describe the usual intake of the population. The changes are highlighted in yellow (Lines 97-128).
Finally, we have included a brief explanation to try to explain the controversial results observed among the different studies. The changes are highlighted in yellow (Lines 889-882).
Lines 904-907: “Nevertheless, while many factors might explain the differences between the studies which failed to demonstrate the possible benefits of the direct action of the bioactive compound of a food consumed, the usual dose of these foods may be very different from the dose actually needed to achieve any beneficial effect.”

Reviewer 3 Report
Lapuente et al. presented a review paper about the key foods in the prevention of atherosclerosis. The review is well-written and well-presented with the details on each of the biomarkers investigated together with percentage magnitude. However, I still have some inputs which might improve the review as follow:
There are different type of fruits and vegetables available showing different beneficial effect for the prevention of atherosclerosis, but what is the reason behind choosing certain food items as presented in this review such as tomatoes, garlic, berries, apples, brocolli, cocoa, grapes and olives ? That would be illustrative to present a figure how those certain fruits/vegetables and its bioactive compounds exert beneficial effects on health. What mechanism behind which need to be supported with in vivo and in vitro studies. That give an improvement in the review to provide an information what bioactive compounds actually contained in each of the presented fruits/vegetables and presented in the table.Author Response
nutrients-581555
RESPONSES TO REVIEWERS:
REVIEWER #3:
Lapuente et al. presented a review paper about the key foods in the prevention of atherosclerosis. The review is well-written and well-presented with the details on each of the biomarkers investigated together with percentage magnitude. However, I still have some inputs which might improve the review as follow:
There are different type of fruits and vegetables available showing different beneficial effect for the prevention of atherosclerosis, but what is the reason behind choosing certain food items as presented in this review such as tomatoes, garlic, berries, apples, brocolli, cocoa, grapes and olives?Comment 1: We decided to focus on the fruits and vegetables showing the greatest evidence of having a role in the prevention of atherosclerosis in the last years. This is justified in lines 139-141.
That would be illustrative to present a figure how those certain fruits/vegetables and its bioactive compounds exert beneficial effects on health.Comment 2: Thank you for your comment. We have included a representative figure (Figure 1) to depict the potential benefits of bioactive compounds on cardiovascular disease.
What mechanism behind which need to be supported with in vivo and in vitro studies. That gives an improvement in the review to provide information what bioactive compounds actually contained in each of the presented fruits/vegetables and presented in the table.Comment 3: Thank you for your comment. We have included a short paragraph to explain the need for further research on the underlying mechanisms through in vitro and in vivo studies because of the controversial results obtained in different studies.The changes are highlighted in yellow (64-68).
Lines 64-68: “While there is sufficient in vitro evidence supporting the antioxidant, anti-inflammatory, and antidiabetic effects, among others, of allicin, flavonoids or stilbenes, the in vivo studies available have reported controversial results, indicating that further research are required. To demonstrate and corroborate the results observed in these in vitro and in vivo studies, large controlled clinical trials are needed in humans [15-19].”

Round 2
Reviewer 2 Report
The Authors have satisfactorily addressed almost all comments.
I still believe that criteria fro excluding articles from the analysis (if any) should be provided.
Polyphenols are not fruit or vegetables. It is not clear to me why this part was added to the review. It may be probably moved at the end of the review as part of a comment on possible mediators of the beneficial effects of fruit and vegetables.
Author Response
nutrients-581555
RESPONSES TO REVIEWERS:
REVIEWER #2:
I still believe that criteria for excluding articles from the analysis (if any) should be provided.
Comment 1: Thank you for your comment. We have added criteria for excluding articles from the analysis. The changes are highlighted in yellow (lines 89-92).
Lines 89-92: “The exclusion criteria consisted in: a) interventional studies published before 2013; b) articles using juices, smoothies or similar products; c) articles not containing some of the characteristics mentioned in inclusion criteria; and d) interventions made in animals, ex-vivo or in-silico.”
Polyphenols are not fruit or vegetables. It is not clear to me why this part was added to the review. It may be probably moved at the end of the review as part of a comment on possible mediators of the beneficial effects of fruit and vegetables.
Comment 2: Thank you very much for your comment again. It is true that polyphenols are not fruits or vegetables. However, they are active compounds that are present in most of them, with wide interesting properties on health. We thought it might be interesting to add something related to their intake.

Reviewer 3 Report
The authors have made changes for the improvement of the paper according to what I have suggested. I'm now satisfied with the the changes and agree to accept the manuscript.
Author Response
The authors have made changes for the improvement of the paper according to what I have suggested. I'm now satisfied with the the changes and agree to accept the manuscript.
Comment: thank you very much